# Design and ARM-Based Implementation of Bitstream-Oriented Chaotic Encryption Scheme for H.264/AVC Video

**DOI:** 10.3390/e23111431

**Published:** 2021-10-29

**Authors:** Zirui Zhang, Ping Chen, Weijun Li, Xiaoming Xiong, Qianxue Wang, Heping Wen, Songbin Liu, Shuting Cai

**Affiliations:** 1School of Automation, Guangdong University of Technology, Guangzhou 510006, China; 2111904292@mail2.gdut.edu.cn (Z.Z.); chenping@gdut.edu.cn (P.C.); xmxiong@gdut.edu.cn (X.X.); wangqianxue@gdut.edu.cn (Q.W.); 2112004118@mail2.gdut.edu.cn (S.L.); 2Zhongshan Institute, University of Electronic Science and Technology of China, Zhongshan 528402, China; wenheping@uestc.edu.cn

**Keywords:** bitstream-oriented encryption, chaotic encryption, format compatibility, H.264/AVC

## Abstract

In actual application scenarios of the real-time video confidential communication, encrypted videos must meet three performance indicators: security, real-time, and format compatibility. To satisfy these requirements, an improved bitstream-oriented encryption (BOE) method based chaotic encryption for H.264/AVC video is proposed. Meanwhile, an ARM-embedded remote real-time video confidential communication system is built for experimental verification in this paper. Firstly, a 4-D self-synchronous chaotic stream cipher algorithm with cosine anti-controllers (4-D SCSCA-CAC) is designed to enhance the security. The algorithm solves the security loopholes of existing self-synchronous chaotic stream cipher algorithms applied to the actual video confidential communication, which can effectively resist the combinational effect of the chosen-ciphertext attack and the divide-and-conquer attack. Secondly, syntax elements of the H.264 bitstream are analyzed in real-time. Motion vector difference (MVD) coefficients and direct-current (DC) components in Residual syntax element are extracted through the Exponential-Golomb decoding operation and entropy decoding operation based on the context-based adaptive variable length coding (CAVLC) mode, respectively. Thirdly, the DC components and MVD coefficients are encrypted by the 4-D SCSCA-CAC, and the encrypted syntax elements are re-encoded to replace the syntax elements of the original H.264 bitstream, keeping the format compatibility. Besides, hardware codecs and multi-core multi-threading technology are employed to improve the real-time performance of the hardware system. Finally, experimental results show that the proposed scheme, with the advantage of high efficiency and flexibility, can fulfill the requirement of security, real-time, and format compatibility simultaneously.

## 1. Introduction

Real-time video communication is widely used in military, business, entertainment, and personal social activities, such as video conferencing, live video, and video surveillance [1,2,3,4]. However, the openness of the internet leads to many security risks in video communication. Cybersecurity incidents such as malicious attacks, illegal access, information leakage, theft, and evil tampering are frequently reported. Many studies have shown that end-to-end encryption is an effective way to protect cyber information and personal privacy [5,6,7,8,9]. Only communicating parties with the right secret keys can easily decrypt ciphertexts, while third parties without the matched secret keys cannot obtain plaintexts even if they obtain the ciphertexts. Therefore, the end-to-end encryption is a necessary means for real-time video confidential communication.

To realize the end-to-end encryption for the communication video, security, real-time, and format compatibility are important technical indicators that must be met simultaneously for the real-time video confidential communication system [1,10,11,12,13,14,15]. Security includes two aspects: cryptographic security and perceptual security. The cryptographic security depends on the ability of encryption algorithms to resist various types of cryptanalysis methods, and the perceptual security relies on the recognizability of encrypted video images from human eyes. Frame rate is considered as the reference index for the real-time performance, and the basic frame rate for real-time video communication is 25 f/s or more. The real-time performance is not only affected by video resolution, but also video compression ratio, calculation load, and computational efficiency [6,12,13]. Format compatibility is another significant indicator of video confidential communication. For instance, the video information of video surveillance, paid video, or live video always needs to be performed adaptation or transcoding operations by third-party servers [5,6,16,17], including storage recognition, watermark insertion, rate adaptation, and so on. Incompatible video format will inevitably lead to the failure of adaptation operations. Although encrypted videos without format compatibility can be decrypted as original videos on the servers before adaptation [5,16], the plaintext information will be exposed on the third-party, which does not meet the requirement of the end-to-end encryption. On the contrary, encrypted videos with format compatibility can be directly adopted without decryption on the servers. Therefore, the compatibility of video format is the primary condition to realize the end-to-end encryption for the communication video [5,18].

To meet the above three indicators simultaneously, in recent years, people perform in-depth studies in encryption algorithms [1,2,10,19,20,21,22,23] and video encryption methods [6,15,16,24,25,26,27,28,29,30,31,32] suitable for real-time video secure communication for H.264/AVC. Using encryption algorithms to encrypt video information in real time is an important means to protect the security of communication videos. However, due to the large amount of data and high relevance of video information in actual video communication applications, traditional text encryption algorithms such as DES and AES, with a large time consumption, are difficult to meet the real-time transmission requirements of encrypted videos [18,33,34]. In the existing encryption algorithms, chaotic encryption algorithm has been extensively explored. Because chaotic encryption algorithm has good pseudo-randomness, initial value sensitivity, and high operation efficiency, it can not only ensure the high security of encrypted video data, but also meet the real-time requirements of video transmission. Therefore, chaotic encryption algorithm is more suitable for real-time video confidential communication applications. In particular, among chaotic encryption algorithms, *n*-D (*n* = 3, 7, 8) self-synchronous chaotic stream cipher algorithms (n-D SCSCA) [19,20,21] applied to an actual video confidential communication system show superior characteristics. These algorithms belong to closed-loop feedback chaotic encryption algorithms. The ciphertext is fed back to the chaotic system to realize the self-synchronization of the encryption and decryption ends. When these algorithms are applied to actual channels, even if the ciphertext information suffers from channel interference during transmission, after the decryption end receives a certain amount of correct ciphertext information, the state variables of the cipher algorithm at both ends can asymptotically synchronize. Thus, these algorithms have a good ability to resist channel interference and are more suitable for actual video confidential communication applications. However, *n*-D SCSCA (*n* = 3, 7, 8) still have security loopholes. Lin et al. [22] propose a cryptanalysis method that combines a chosen-ciphertext attack with a divide-and-conquer attack by traversing single non-zero component initial conditions (DCA-TSNCIC). The method is used to decipher 8-D SCSCA [19] that employ the lower eight-bits of a single chaotic variable as the chaotic pseudo-random sequence, and most of the secret keys of 8-D SCSCA are deciphered. To improve the security of SCSCA, refs [20,21] propose *n*-D SCSCA (*n* = 3, 7) that employ the lower eight-bits of the multiplication result of multiple chaotic variables as the chaotic pseudo-random sequence. *n*-D SCSCA (*n* = 3, 7) increase the computational complexity of the chaotic pseudo-random sequence expression, so that cryptanalysts cannot obtain enough nonlinear equations through DCA-TSNCIC to solve the original secret keys of the encryption algorithm. However, they cannot further resist the divide-and-conquer attack traversing multiple non-zero component initial conditions (DCA-TMNCIC) that is more powerful than the DCA-TSNCIC [35].

In the existing video encryption scheme for H.264/AVC, they can be classified into full encryption (FE) [32] and selective encryption (SE) [16,30,31,32]. The FE method is to indiscriminately encrypt the entire encoded video information, which will cause the format information of the encoded video to be destroyed [17]. The SE method is to selectively encrypt part of the critical information in the encoded video, and the selected critical information will not affect the encoded video format after being encrypted. Compared with the FE method, the SE method has less encrypted data and can maintain the format of encrypted video. Thus, the SE scheme is an effective method to realize the security, real-time, and format compatibility of the video confidential communication system.

According to the different relationships between encryption algorithms and video codecs, SE methods can be further subdivided into compression-integrated encryption (CIE) and bitstream-oriented encryption (BOE). The feature of CIE is that it has a coupling structure embedding the encryption algorithm into the video codec, where the video information is encrypted during encoding. The feature of BOE is that it has an independent architecture separating the encryption algorithm and the video codec, where the video information is encrypted after being encoded. Chen et al. [32] and Zhang et al. [36] designed a video confidential communication system based on the CIE scheme. The encryption algorithm was embedded into software codecs, and the syntax elements in the H.264 bitstream were encrypted during the encoding process. The experimental results prove that the CIE scheme can realize that encrypted videos keep the H.264 format, fulfilling the format compatibility. However, under the 640 × 480 video resolution, the video transmission frame rate in [32,36] cannot meet the basic real-time requirement (25 f/s) in the actual video communication. The main reason for this result is that software codecs with low computational efficiency seriously damage real-time performance. Besides, the CIE scheme lacks flexibility owing to the dependency between the encryption algorithm and the software codecs. Any standard codecs, no matter the software or hardware, are not applicable in the CIE scheme. Only the special software codecs customized with the encryption algorithm are suitable for it. Compared with the CIE scheme, the BOE scheme is more flexible than the CIE scheme because any standard codecs can be used to perform the coding operation in the independent architecture. Arachchi et al. [5] proposed a standalone encryption method to achieve end-to-end security adaptation requirements. The standalone encryption method separated the video encoding and video encryption process. Meanwhile, it performed a series operation of analysis, extraction, encryption, and encoding on the H.264 bitstream, so it can be classified as the BOE method. References [3,18,23] proposed a similar encryption method for H.264 bitstream, which retained the ‘header’ information of the H.264 bitstream and encrypted the rest of the data. When the type of ‘header’ is revised as unspecific, the encrypted bitstream will be bypassed without decoding to maintain format compatibility. Although this method performed encryption operation for the H.264 bitstream with an independent architecture of the encryption algorithm and codec, it is still far from the BOE method. In fact, the method is more like FE than SE due to the main information of the encoded videos being encrypted indiscriminately. Its biggest flaw comes from the destruction on the syntactic structure of macroblocks. Boyadjis et al. [37] and Cheng et al. [4] encrypted syntax elements extracted from H.264 bitstream and generated encrypted video with format compatibility. They all adopted the combination of BOE method and AES block cipher to design a video encryption system. AES algorithm has to construct a fixed-length packet for encryption and even fill up the package with padding data when the packet data is insufficient. The data filling operation will increase computational load and the amount of encrypted data. Moreover, the encryption test objects are static video files rather than real-time video streaming media. The experiment is conducted by simulation on PC rather than by real test on the embedded hardware platform under the actual network environment. Therefore, they do not provide strong proof of feasibility, effectiveness, and superiority for the BOE method. To summarize, the comparison of the features of BOE, CIE, and FE methods are given in Table 1.

To address the above issues, for H.264/AVC video, an improved BOE is proposed and verified in an actual network environment upon the ARM-embedded hardware platform. The main contributions and novelties of this work can be summarized as follows:

(1) A 4-D SCSCA-CAC is proposed, which can effectively resist the cryptanalysis method combining a chosen-ciphertext attack with a divide-and-conquer attack. In addition, the chaotic bit sequences generated by 4-D SCSCA-CAC for encrypting video information have passed the NIST and TESTU01 test.

(2) An improved BOE method based on chaotic stream ciphers is proposed, which separates apart the hardware codec and 4-D SCSCA-CAC to form an independent architecture. The proposed scheme can achieve the format compatibility of encrypted videos and balance the technical contradiction between security and real-time performance.

(3) An ARM-based hardware system for real-time video confidential communication is designed and implemented, whose experimental results give a practical and objective evaluation for the critical issues about security, real-time, and format compatibility, verifying feasibility and effectiveness for the BOE method.

The rest of the paper is organized as follows: Section 2 introduces the design of the improved chaotic stream cipher and its security analysis. Section 3 describes the design and ARM-based implementation of the bitstream-oriented chaotic encryption scheme. Section 4 presents the experimental results and test analysis. Section 5 concludes the paper.

## 2. Design and Security Analysis of Chaotic Stream Cipher Algorithm

### 2.1. Security Loopholes Analysis of n-D SCSCA

In order to solve the above security problem of *n*-D SCSCA (*n* = 3, 7, 8) [19,20,21], one first analyzes their security loopholes, and designs 4-D SCSCA-CAC with great security.

According to the structure characteristics of *n*-D SCSCA (*n* = 3, 7, 8), and combining with the cryptanalysis process of chosen-ciphertext attack and divide-and-conquer attack, the security loopholes of *n*-D SCSCA (*n* = 3, 7, 8) are analyzed in this subsection.

The algorithm structure of *n*-D SCSCA (*n* = 3, 7, 8) and corresponding cryptanalysis methods are shown in Table 2.

From Table 2, the general form of *n*-D SCSCA (*n* = 3,7,8) can be summarized as:(1)$$x\left(k+1\right)=f\left(a_{ij},x\left(k\right),p\left(k\right)\right)+g\left(\sigma_{l}p\left(k\right),\varepsilon_{l}\right)\;,$$
where k=0,1,2,3,⋯, p(k) represents the ciphertext, $f(a_{ij},x\left(k\right),p\left(k\right))$ represents a nominal system adopting ciphertext feedback, $g(\sigma_{l}p\left(k\right),\varepsilon_{l})$ represents the uniformly bounded controller. $a_{ij},\sigma_{l},\varepsilon_{l}\;\left(i,j=3,7,8;l=1,3\right)$ represents the secret keys, $x\left(k+1\right)=(x_{1}\left(k+1\right),x_{2}\left(k+1\right),\cdots ,x_{n}(k$$+1){)}^{T}$ and $x\left(k\right)={\left(x_{1}\left(k\right),x_{2}\left(k\right),\cdots ,x_{n}\left(k\right)\right)}^{T}\;\left(n=3,7,8\right)$ represents chaotic variables.

The general form of decryption expression can be summarized as follows:(2)$$m\left(k\right)=s\left(k\right)⊕p\left(k\right)=\;\mathrm{mod}\;\left(\left\lfloor x_{i}\left(k\right)\cdots x_{i+j}\left(k\right)/2^{w}\right\rfloor ,2^{8}\right)⊕p\left(k\right)\;,$$
where k=0,1,2,3,⋯, 1≤i≤8, w≤27, 1≤j≤(8−i), $\left\lfloor \cdot \right\rfloor$ represents round-down operation, ⊕ represents the bitwise XOR operation, m(k) represents the plaintext, s(k) represents the chaotic pseudo-random sequence.

Note that in Table 2, to achieve self-synchronization, the ciphertext is used as a feedback control variable of the anti-controller. Therefore, under the condition of the chosen-ciphertext attack, the cryptanalyst can select the specific ciphertext and feed it back into the anti-controllers, to affect the calculation result of the anti-controllers. Moreover, the anti-controller types of *n*-D SCSCA (*n* = 3,7,8) are mod and sine function. When the ciphertext is set as p(k)=0, the calculation result of the anti-controllers will be zero. This situation will cause the original nonlinear iterative equation to degenerate into a linear iterative equation. According to Equation (Equation 1), setting the ciphertext as p(k)=0, a linear iterative equation is derived as:(3)$$x\left(k+1\right)=f(a_{ij},x\left(k\right))\;.$$

Compared Equation (Equation 3) with Equation (Equation 1), the calculation complexity of Equation (Equation 3) is greatly reduced and the anti-controller secret keys $\sigma_{l},\varepsilon_{l}\;\left(l=1,3\right)$ are eliminated, which provides an important precondition for using the divide-and-conquer attack to decipher the secret keys in *n*-D SCSCA (*n* = 3, 7, 8).

In SCSCA, initial conditions of arbitrary chaotic variables at the decryption end can achieve asymptotic synchronization with the decryption end, hence the cryptanalyst can arbitrarily select the initial conditions conducive to cryptanalysis. In the case of chosen-ciphertext attack, Equation (Equation 2) can be simplified as:(4)$$m\left(k\right)=s\left(k\right)⊕p\left(k\right)=s\left(k\right)⊕0=s\left(k\right)=\;\mathrm{mod}\;\left(\left\lfloor F^{\left(k\right)}\left(a_{ij},c\right)\right\rfloor ,2^{8}\right)\;,$$
where k=0,1,2,3,⋯, $c=(x_{1}\left(0\right),x_{2}\left(0\right),\cdots ,x_{n}\left(0\right))$, $F^{\left(k\right)}\left(a_{ij},c\right)$ represents the relational expression between $a_{ij}$ and *c*.

Next, according to the different computational complexity of the chaotic pseudo-random sequence s(k) in *n*-D SCSCA (*n* = 3,7,8), the secret keys can be cracked through the divide-and-conquer attack methods with different strengths, respectively. When s(k) is generated by intercepting the lower eight-bits of a single chaotic variable, the encryption algorithm can be cracked by DCA-TSNCIC [22]. With DCA-TSNCIC, the set of *n* selection methods of initial conditions is as follow:(5)$$\begin{array}{c} \left\{c\right\}=\left\{\left(x_{1}\left(0\right),x_{2}\left(0\right),\cdots ,x_{n}\left(0\right)\right)\right\}=\{\left(c_{1},0,\cdots ,0\right),\left(0,c_{2},0,\cdots ,0\right),\\ \;\;\;\;\;\cdots ,\left(0,\cdots ,0,c_{n-1},0\right),\left(0,0,\cdots ,c_{n}\right)\}, \end{array}$$
where $c_{i}\;\left(i=1,2,\cdots ,n\right)$ represents non-zero constants.

When s(k) is generated by intercepting the lower eight-bits of the multiplication result of multiple chaotic variables, the encryption algorithm can be deciphered by DCA-TMNCIC [35] which has a higher attack intensity than DCA-TSNCIC [22]. With DCA-TMNCIC, the set of $2^{n}-1$ selection methods of initial conditions is as follow:(6)$$\begin{array}{c} \left\{c\right\}=\left\{\left(x_{1}\left(0\right),x_{2}\left(0\right),\cdots ,x_{n}\left(0\right)\right)\right\}=\{\left(c_{1},0,\cdots ,0\right),\left(0,c_{2},0,\cdots ,0\right),\\ \;\;\cdots \left(0,0,\cdots ,c_{n}\right),\left(c_{1},c_{2},0,\cdots ,0\right),\cdots ,\left(c_{1},c_{2},\cdots ,c_{n-1},c_{n}\right)\}, \end{array}$$

Remarkably, in *n*-D SCSCA (*n* = 3,7,8), due to s(k) is generated by intercepting the lower eight-bits of a single chaotic variable or the multiplication result of multiple chaotic variables, when values of initial conditions are set as a same non-zero constant $c_{i}=c_{0}\;\left(i=1,2,\cdots ,n\right)$, a constant common factor $c_{0}^{m}\;\left(m=1,2,\cdots \right)$ multiplied with secret key expressions can be extracted from the decryption expression. Therefore, Equation (Equation 4) can be further simplified as:(7)$$m\left(k\right)=\;\mathrm{mod}\;\left(\left\lfloor c_{0}^{m}f\left(a_{ij}\right)\right\rfloor ,2^{8}\right)\;,$$
where $f(a_{ij})$ represents the secret key expression.

The principle of the divide-and-conquer attack is to decipher the information of secret key or secret key expression block by block. In Equation (Equation 7), $f(a_{ij})$ is represented by the 64-bit binary numbers, which is denoted by ${\left(f\left(a_{ij}\right)\right)}_{2}$. The first bit in ${\left(f\left(a_{ij}\right)\right)}_{2}$ represents the sign bit, and the remaining 63-bits represents the data. According to the divide-and-conquer attack, ${\left(f\left(a_{ij}\right)\right)}_{2}$ are divided into 8 sub-blocks, respectively. Each sub-block is of 8-bits, denoted by ${\left(f{\left(a_{ij}\right)}^{\left(k\right)}\right)}_{2}\;\left(k=1,2,\cdots ,8\right)$. Finally, the cryptanalyst can sequentially set $c_{0}$ as $\sqrt[{m}]{2^{7+8i}}\;\left(i=0,1,2,\cdots 7\right)$ to obtain the information of each sub-block ${\left(f{\left(a_{ij}\right)}^{\left(1\right)}\right)}_{2},{\left(f{\left(a_{ij}\right)}^{\left(2\right)}\right)}_{2},\cdots ,{\left(f{\left(a_{ij}\right)}^{\left(8\right)}\right)}_{2}$, a complete secret key expression information can be derived as:(8)$$\begin{array}{c} \left(f{\left(a_{ij}\right)}_{2}\right)={\left(f{\left(a_{ij}\right)}^{\left(1\right)}\right)}_{2}{\left(f{\left(a_{ij}\right)}^{\left(2\right)}\right)}_{2}{\left(f{\left(a_{ij}\right)}^{\left(3\right)}\right)}_{2}{\left(f{\left(a_{ij}\right)}^{\left(4\right)}\right)}_{2}\\ \;\;{\left(f{\left(a_{ij}\right)}^{\left(5\right)}\right)}_{2}{\left(f{\left(a_{ij}\right)}^{\left(6\right)}\right)}_{2}{\left(f{\left(a_{ij}\right)}^{\left(7\right)}\right)}_{2}{\left(f{\left(a_{ij}\right)}^{\left(8\right)}\right)}_{2}. \end{array}$$
where $f(a_{ij})$ represents the secret key expression.

From the above method, a sufficient number of nonlinear equations about secret keys can be obtained, the correct information of the original secret keys can be deciphered by solving the nonlinear equations.

According to the above analysis, the main problems of *n*-D SCSCA (*n* = 3,7,8) can be summarized as follows:

(1) To achieve self-synchronization, the ciphertext need to be fed back into the anti-controllers of the chaotic system, and the anti-controller types of *n*-D SCSCA (*n* = 3,7,8) are sine or mod function. In this case, when the cryptanalyst set ciphertext information as p(k)=0 by the chosen-ciphertext attack, the chaotic system will be degenerated into a linear system, and the controller secret keys are eliminated.

(2) In actual channel communications, initial conditions of arbitrary chaotic variables at the receiver can achieve asymptotic synchronization with the sender. Therefore, the initial conditions of the chaotic variables belong to the weak secret keys, and the cryptanalyst can arbitrarily select the initial conditions conducive to cryptanalysis. Besides, *n*-D SCSCA (*n* = 3,7,8) all use the lower eight-bits of a single chaotic variable or the multiplication result of multiple chaotic variables as the chaotic pseudo-random sequence. With the chosen-ciphertext attack, when all initial conditions are set as a same non-zero constant, a constant common factor can be extracted from the decryption expression. Consequently, by setting the appropriate value of the constant, the cryptanalyst can use the divide-and-conquer attack to obtain the complete information of the secret key expression. After acquiring a sufficient number of nonlinear equations about secret keys, the original secret keys can be successfully deciphered through solving the nonlinear equations.

### 2.2. Design of 4-D Chaotic Cipher Algorithm

To deal with the loopholes of SCSCA analyzed in Section 2.1, in this paper, one proposes an improved scheme named 4-D SCSCA-CAC, which can effectively against the combination of the chosen-ciphertext attack and the divide-and-conquer attack.


**A.**
*
**Design An Asymptotically Stable Nominal System**
*


An uncontrolled 4-D discrete-time linear nominal system is expressed as:(9)x(k+1)=Ax(k),
where
(10)$$x\left(k+1\right)=\left(\begin{array}{c} x_{1}\left(k+1\right)\\ x_{2}\left(k+1\right)\\ x_{3}\left(k+1\right)\\ x_{4}\left(k+1\right) \end{array}\right),\;x\left(k\right)=\left(\begin{array}{c} x_{1}\left(k\right)\\ x_{2}\left(k\right)\\ x_{3}\left(k\right)\\ x_{4}\left(k\right) \end{array}\right),A={\left[\begin{array}{cccc} a_{11} & a_{12} & a_{13} & a_{14}\\ a_{21} & a_{22} & a_{23} & a_{24}\\ a_{31} & a_{32} & a_{33} & a_{34}\\ a_{41} & a_{42} & a_{43} & a_{44} \end{array}\right]}_{4\times 4},$$

According to the design principle of discrete-time chaotic system [38], all eigenvalues of matrix A must be located inside the unit circle on the complex plane to make the nominal system asymptotically stable.

First, the MATLAB rand() function is used to generate a 4-D full rank matrix with uncorrelated random values in the range of (−1, 1). Next, a orthonormal matrix Q is obtained by orthonormalizing the 4-D matrix using the orth() function. Finally, since all eigenvalues of orthonormal matrix are located on the unit circle on the complex plane, when matrix Q is multiplied by a coefficient less than 1, a matrix A with all eigenvalues located inside the unit circle on the complex plane can be obtained, given by:(11)$$A={\left[\begin{array}{cccc} -0.02403 & -0.48546 & -0.61686 & -0.43956\\ -0.55197 & -0.51480 & -0.03528 & -0.48897\\ 0.77443 & -0.32805 & 0.03681 & -0.45252\\ -0.09216 & -0.44901 & -0.65340 & -0.41586 \end{array}\right]}_{4\times 4}.$$

From Equation (Equation 11), one gets four characteristic roots of the matrix A in the nominal system (Equation 9), that are $\lambda_{1}=0.9$, $\lambda_{2}=0.0558+j0.8983$, $\lambda_{3}=0.0558-j0.8983$, $\lambda_{4}=-0.9$. Thus, the nominal system (Equation 9) is asymptotically stable.


**B.**
*
**The 4-D Controlled System Based Anti-control Principle**
*


According to the principle of anti-control of dynamical systems [38], selecting $x_{1}\left(k\right)$ as feedback control variable, three uniformly bounded feedback controllers are designed as:(12)$$\varepsilon_{i}\cos\left(\sigma_{j}x_{1}\left(k\right)\right)\left(i,j=1,2,3\right),$$
where $\varepsilon_{1}=2.2\times 10^{10}$, $\varepsilon_{2}=2.4\times 10^{10}$, $\varepsilon_{3}=2.6\times 10^{10}$, $\sigma_{1}=\sigma_{2}=\sigma_{3}=7\times 10^{8}$. Then, by applying the three feedback controllers to the second and third equations of the nominal system (Equation 9), the 4-D controlled system is obtained as:(13)$$\left\{\begin{array}{c} x_{1}\left(k+1\right)=a_{11}x_{1}\left(k\right)+a_{12}x_{2}\left(k\right)+a_{13}x_{3}\left(k\right)+a_{14}x_{4}\left(k\right)\\ x_{2}\left(k+1\right)=a_{21}\varepsilon_{1}\cos\left(\sigma_{1}x_{1}\left(k\right)\right)+a_{22}x_{2}\left(k\right)+a_{23}x_{3}\left(k\right)+a_{24}x_{4}\left(k\right)\\ x_{3}\left(k+1\right)=a_{31}\varepsilon_{2}\cos\left(\sigma_{2}x_{1}\left(k\right)\right)+a_{32}x_{2}\left(k\right)+a_{33}x_{3}\left(k\right)+a_{34}x_{4}\left(k\right)\\ x_{4}\left(k+1\right)=a_{41}\varepsilon_{3}\cos\left(\sigma_{3}x_{1}\left(k\right)\right)+a_{42}x_{2}\left(k\right)+a_{43}x_{3}\left(k\right)+a_{44}x_{4}\left(k\right) \end{array}\right.$$

The corresponding Lyapunov exponents of the system (Equation 13) obtained by simulation calculation are $LE_{1}=22.25$, $LE_{2}=20.85$, $LE_{3}=4.98$, $LE_{4}=3.49$. According to Theorem 1 [38], the controlled system (Equation 13) is chaotic, and the corresponding chaos attractor phase diagram is shown in Figure 1:


**C.**
*
**4-D SCSCA-CAC**
*


Note that, since application environment of the chaotic system is ARM-embedded platform in this paper, the following calculation operations are based on binary representation. The chaotic pseudo-random sequence s(k) used for encryption operation is expressed as:(14)$$s\left(k\right)=\;\mathrm{mod}\;\left(\left\lfloor \frac{x_{1}\left(k\right)\times x_{2}\left(k\right)+x_{3}\left(k\right)}{2^{16}}\right\rfloor ,2^{8}\right),$$
where $\left\lfloor \cdot \right\rfloor$ represents round-down operation, $\;\mathrm{mod}\;(\cdot ,2^{8})$ is used for intercepting the lower eight-bits on $\left\lfloor \left(x_{1}\left(k\right)\times x_{2}\left(k\right)+x_{3}\left(k\right)\right)/2^{16}\right\rfloor$ to generate eight-bits chaotic pseudo-random sequence s(k).

Then, the ciphertext p(k) can be expressed as:(15)$$p\left(k\right)=s\left(k\right)⊕m\left(k\right)\;\mathrm{mod}\;\left(\left\lfloor \frac{x_{1}\left(k\right)\times x_{2}\left(k\right)+x_{3}\left(k\right)}{2^{16}}\right\rfloor ,2^{8}\right)⊕m\left(k\right),$$
where m(k) represents the plaintext information, ⊕ represents the bitwise XOR operation.

To realize self-synchronization, the ciphertext p(k) needs to be fed back and substituted for the chaotic system to participate in the iterative calculation [19,20,21], then 4-D SCSCA-CAC is designed as:(16)$$\left\{\begin{array}{c} x_{1}\left(k+1\right)=a_{11}x_{1}\left(k\right)+a_{12}x_{2}\left(k\right)+a_{13}x_{3}\left(k\right)+a_{14}x_{4}\left(k\right)\\ x_{2}\left(k+1\right)=a_{21}\varepsilon_{1}\cos\left(\sigma_{1}p\left(k\right)\right)+a_{22}x_{2}\left(k\right)+a_{23}x_{3}\left(k\right)+a_{24}x_{4}\left(k\right)\\ x_{3}\left(k+1\right)=a_{31}\varepsilon_{2}\cos\left(\sigma_{2}p\left(k\right)\right)+a_{32}x_{2}\left(k\right)+a_{33}x_{3}\left(k\right)+a_{34}x_{4}\left(k\right)\\ x_{4}\left(k+1\right)=a_{41}\varepsilon_{3}\cos\left(\sigma_{3}p\left(k\right)\right)+a_{42}x_{2}\left(k\right)+a_{43}x_{3}\left(k\right)+a_{44}x_{4}\left(k\right) \end{array}\right.$$

The main features of 4-D SCSCA-CAC are summarized as follows:

(1) Unlike the anti-controllers shown in Table 2, a cosine function is used as the anti-controller. With p(k)=0, the controller parameters remain in the equation without being eliminated, which can increase the difficulty of the key parameters deciphering, especially increasing the ability to resist the divide-and-conquer attack. Note that $\sigma_{i},\varepsilon_{i}\neq 0\;\left(i=1,2,3\right)$ in Equation (Equation 16), consequently $\varepsilon_{i}\cos\left(\sigma_{i}p\left(k\right)\right)\neq 0$ within the value range of {p(k)|0≤p(k)≤255,p(k)∈N}. Although Equation (Equation 16) degrades into a linear iterative equation with p(k)=0, $\varepsilon_{i}\;$ can still retain in the linear iterative equation and provide a powerful condition to resist the divide-and-conquer attack.

(2) Note that an additive term is introduced in the modulo operation and the round-down operation in Equation (Equation 14); resulting m(k) contains addition independent terms that do not multiply by the initial conditions, under the divide-and-conquer attack.

### 2.3. Security Analysis

From the chosen-ciphertext attack method, it can be known that when the security analysis of 4-D SCSCA-CAC is performed, the cryptanalyst can arbitrarily select the ciphertext that is beneficial to the decryption algorithm and obtain the corresponding plaintext. When the ciphertext is set as p(k)=0, the mathematical expression of the linear iterative equation can be derived as:(17)$$\left\{\begin{array}{c} x_{1}\left(k+1\right)=a_{11}x_{1}\left(k\right)+a_{12}x_{2}\left(k\right)+a_{13}x_{3}\left(k\right)+a_{14}x_{4}\left(k\right)\\ x_{2}\left(k+1\right)=a_{21}\varepsilon_{1}+a_{22}x_{2}\left(k\right)+a_{23}x_{3}\left(k\right)+a_{24}x_{4}\left(k\right)\\ x_{3}\left(k+1\right)=a_{31}\varepsilon_{2}+a_{32}x_{2}\left(k\right)+a_{33}x_{3}\left(k\right)+a_{34}x_{4}\left(k\right)\\ x_{4}\left(k+1\right)=a_{41}\varepsilon_{3}+a_{42}x_{2}\left(k\right)+a_{43}x_{3}\left(k\right)+a_{44}x_{4}\left(k\right) \end{array}\right.$$
where k=0,1,2,⋯. By substituting p(k)=0 into Equation (Equation 15), which yields:(18)$$m\left(k\right)\;=\;\mathrm{mod}\;\left(\left\lfloor \frac{x_{1}\left(k\right)\times x_{2}\left(k\right)+x_{3}\left(k\right)}{2^{16}}\right\rfloor ,2^{8}\right)$$

From Equation (Equation 17), one can see that although the chaotic system is degenerated into a linear iterative equation with p(k)=0, anti-controller secret keys $\varepsilon_{i}\;\left(i=1,2,3\right)$ still retain in the linear iterative equation and will provide a powerful condition to resist divide-and-conquer attack.

In the decryption process, decryption end can achieve asymptotic synchronization with the encryption end under any given initial conditions, so the cryptanalyst can choose any initial conditions that are conducive to deciphering the encryption algorithm. In summary, when the ciphertext p(k)=0 is set in the 4-D SCSCA-CAC, the initial condition value $x_{i}\left(0\right)=c_{i}\left(i=1,2,3,4\right)$ can be arbitrarily selected to try to obtain the secret keys information by the divide-and-conquer attack. Next, DCA-TMNCIC with higher attack strength than DCA-TSNCIC is used to analyze the security of 4-D SCSCA-CAC.

Firstly, by substituting k=0 into Equation (Equation 17), the first iteration result is given by:(19)$$\left\{\begin{array}{c} x_{1}\left(1\right)=a_{11}x_{1}\left(0\right)+a_{12}x_{2}\left(0\right)+a_{13}x_{3}\left(0\right)+a_{14}x_{4}\left(0\right)\\ x_{2}\left(1\right)=a_{21}\varepsilon_{1}+a_{22}x_{2}\left(0\right)+a_{23}x_{3}\left(0\right)+a_{24}x_{4}\left(0\right)\\ x_{3}\left(1\right)=a_{31}\varepsilon_{2}+a_{32}x_{2}\left(0\right)+a_{33}x_{3}\left(0\right)+a_{34}x_{4}\left(0\right)\\ x_{4}\left(1\right)=a_{41}\varepsilon_{3}+a_{42}x_{2}\left(0\right)+a_{43}x_{3}\left(0\right)+a_{44}x_{4}\left(0\right) \end{array}\right.$$

Then, by substituting k=1 into Equation (Equation 18), the second decryption operation result is given by:(20)$$m\left(1\right)\;=\;\mathrm{mod}\;\left(\left\lfloor \frac{x_{1}\left(1\right)\times x_{2}\left(1\right)+x_{3}\left(1\right)}{2^{16}}\right\rfloor ,2^{8}\right)$$

With DCA-TMNCIC for 4-D SCSCA-CAC, the set of fifteen selection methods of initial conditions is as follow:(21)$$\begin{array}{c} \left(x_{1}\left(0\right),x_{2}\left(0\right),x_{3}\left(0\right),x_{4}\left(0\right)\right)\in \{\left(c_{1},0,0,0\right),\left(0,c_{2},0,0\right),\left(0,0,c_{3},0\right),\left(0,0,0,c_{4}\right),\\ \;\left(c_{1},c_{2},0,0\right),\left(c_{1},0,c_{3},0\right),\left(c_{1},0,0,c_{4}\right),\left(0,c_{2},c_{3},0\right),\left(0,c_{2},0,c_{4}\right),\left(0,0,c_{3},c_{4}\right),\\ \;\left(c_{1},c_{2},c_{3},0\right),\left(c_{1},c_{2},0,c_{4}\right),\left(c_{1},0,c_{3},c_{4}\right),\left(0,c_{2},c_{3},c_{4}\right),\left(c_{1},c_{2},c_{3},c_{4}\right)\} \end{array}$$

Note that, in the case of choosing the ciphertext, the ciphertexts and corresponding plaintexts are both known. Therefore, by substituting the fifteen initial conditions in Equation (Equation 21) into equation Equation (Equation 19), one can obtain $m_{i}\left(1\right)\left(i=1,2,\cdots ,15\right)$ from Equation (Equation 20) as follows:
(22)$$\left\{\begin{array}{c} m_{1}\left(1\right)=\;\mathrm{mod}\;\left(\left\lfloor \frac{a_{11}a_{21}\varepsilon_{1}c_{1}+a_{31}\varepsilon_{2}}{2^{16}}\right\rfloor ,2^{8}\right)\\ m_{2}\left(1\right)=\;\mathrm{mod}\;\left(\left\lfloor \frac{\left(a_{12}a_{22}c_{2}+a_{12}a_{21}\varepsilon_{1}+a_{32}\right)c_{2}+a_{31}\varepsilon_{2}}{2^{16}}\right\rfloor ,2^{8}\right)\\ m_{3}\left(1\right)=\;\mathrm{mod}\;\left(\left\lfloor \frac{\left(a_{13}a_{23}c_{3}+a_{13}a_{21}\varepsilon_{1}+a_{33}\right)c_{3}+a_{31}\varepsilon_{2}}{2^{16}}\right\rfloor ,2^{8}\right)\\ m_{4}\left(1\right)=\;\mathrm{mod}\;\left(\left\lfloor \frac{\left(a_{14}a_{24}c_{4}+a_{14}a_{21}\varepsilon_{1}+a_{34}\right)c_{4}+a_{31}\varepsilon_{2}}{2^{16}}\right\rfloor ,2^{8}\right)\\ m_{5}\left(1\right)=\;\mathrm{mod}\;\left(\left\lfloor \frac{\left(a_{11}c_{1}+a_{12}c_{2}\right)\left(a_{22}c_{2}+a_{21}\varepsilon_{1}\right)+a_{32}c_{2}+a_{31}\varepsilon_{2}}{2^{16}}\right\rfloor ,2^{8}\right)\\ m_{6}\left(1\right)=\;\mathrm{mod}\;\left(\left\lfloor \frac{\left(a_{11}c_{1}+a_{13}c_{3}\right)\left(a_{23}c_{3}+a_{21}\varepsilon_{1}\right)+a_{33}c_{3}+a_{31}\varepsilon_{2}}{2^{16}}\right\rfloor ,2^{8}\right)\\ m_{7}\left(1\right)=\;\mathrm{mod}\;\left(\left\lfloor \frac{\left(a_{11}c_{1}+a_{14}c_{4}\right)\left(a_{24}c_{4}+a_{21}\varepsilon_{1}\right)+a_{34}c_{3}+a_{31}\varepsilon_{2}}{2^{16}}\right\rfloor ,2^{8}\right)\\ m_{8}\left(1\right)=\;\mathrm{mod}\;\left(\left\lfloor \frac{\left(a_{12}c_{2}+a_{13}c_{3}\right)\left(a_{22}c_{2}+a_{23}c_{3}+a_{21}\varepsilon_{1}\right)+a_{32}c_{2}+a_{33}c_{3}+a_{31}\varepsilon_{2}}{2^{16}}\right\rfloor ,2^{8}\right)\;\;\;\;\;\;\;\\ m_{9}\left(1\right)=\;\mathrm{mod}\;\left(\left\lfloor \frac{\left(a_{12}c_{2}+a_{14}c_{4}\right)\left(a_{22}c_{2}+a_{24}c_{4}+a_{21}\varepsilon_{1}\right)+a_{32}c_{2}+a_{34}c_{4}+a_{31}\varepsilon_{2}}{2^{16}}\right\rfloor ,2^{8}\right)\\ m_{10}\left(1\right)=\;\mathrm{mod}\;\left(\left\lfloor \frac{\left(a_{13}c_{3}+a_{14}c_{4}\right)\left(a_{23}c_{3}+a_{24}c_{4}+a_{21}\varepsilon_{1}\right)+a_{33}c_{3}+a_{34}c_{4}+a_{31}\varepsilon_{2}}{2^{16}}\right\rfloor ,2^{8}\right)\\ m_{11}\left(1\right)=\;\mathrm{mod}\;\left(\left\lfloor \frac{\left(a_{11}c_{1}+a_{12}c_{2}+a_{13}c_{3}\right)\left(a_{22}c_{2}+a_{23}c_{3}+a_{21}\varepsilon_{1}\right)+a_{32}c_{2}+a_{33}c_{3}+a_{31}\varepsilon_{2}}{2^{16}}\right\rfloor ,2^{8}\right)\\ m_{12}\left(1\right)=\;\mathrm{mod}\;\left(\left\lfloor \frac{\left(a_{11}c_{1}+a_{12}c_{2}+a_{14}c_{4}\right)\left(a_{22}c_{2}+a_{24}c_{4}+a_{21}\varepsilon_{1}\right)+a_{32}c_{2}+a_{34}c_{4}+a_{31}\varepsilon_{2}}{2^{16}}\right\rfloor ,2^{8}\right)\\ m_{13}\left(1\right)=\;\mathrm{mod}\;\left(\left\lfloor \frac{\left(a_{11}c_{1}+a_{13}c_{3}+a_{14}c_{4}\right)\left(a_{23}c_{3}+a_{24}c_{4}+a_{21}\varepsilon_{1}\right)+a_{33}c_{3}+a_{34}c_{4}+a_{31}\varepsilon_{2}}{2^{16}}\right\rfloor ,2^{8}\right)\\ m_{14}\left(1\right)=\;\mathrm{mod}\;\left(\left\lfloor \frac{\left(a_{12}c_{2}+a_{13}c_{3}+a_{14}c_{4}\right)\left(a_{22}c_{2}+a_{23}c_{3}+a_{24}c_{4}+a_{21}\varepsilon_{1}\right)+a_{32}c_{2}+a_{33}c_{3}+a_{34}c_{4}+a_{31}\varepsilon_{2}}{2^{16}}\right\rfloor ,2^{8}\right)\\ m_{15}\left(1\right)=\;\mathrm{mod}\;\left(\left\lfloor \frac{\left(a_{11}c_{1}+a_{12}c_{2}+a_{13}c_{3}+a_{14}c_{4}\right)\left(a_{22}c_{2}+a_{23}c_{3}+a_{24}c_{4}+a_{21}\varepsilon_{1}\right)+a_{32}c_{2}+a_{33}c_{3}+a_{34}c_{4}+a_{31}\varepsilon_{2}}{2^{16}}\right\rfloor ,2^{8}\right) \end{array}\right.$$

Then, by setting the initial conditions as a same non-zero constant $c_{i}=c_{0}\;\left(i=1,2,3,4\right)$ into Equation (Equation 22), one gets:
(23)$$\left\{\begin{array}{c} m_{1}\left(1\right)=\;\mathrm{mod}\;\left(\left\lfloor \frac{\begin{array}{c} a_{11}a_{21}\varepsilon_{1}c_{0}\\ +a_{31}\varepsilon_{2} \end{array}}{2^{16}}\right\rfloor ,2^{8}\right)\\ m_{2}\left(1\right)=\;\mathrm{mod}\;\left(\left\lfloor \frac{\begin{array}{c} a_{12}a_{22}c_{0}^{2}+(a_{12}a_{21}\varepsilon_{1}\\ +a_{32})c_{0}+a_{31}\varepsilon_{2} \end{array}}{2^{16}}\right\rfloor ,2^{8}\right)\\ m_{3}\left(1\right)=\;\mathrm{mod}\;\left(\left\lfloor \frac{\begin{array}{c} a_{13}a_{23}c_{0}^{2}+(a_{13}a_{21}\varepsilon_{1}\\ +a_{33})c_{0}+a_{31}\varepsilon_{2} \end{array}}{2^{16}}\right\rfloor ,2^{8}\right)\\ m_{4}\left(1\right)=\;\mathrm{mod}\;\left(\left\lfloor \frac{\begin{array}{c} a_{14}a_{24}c_{0}^{2}+(a_{14}a_{21}\varepsilon_{1}\\ +a_{34})c_{0}+a_{31}\varepsilon_{2} \end{array}}{2^{16}}\right\rfloor ,2^{8}\right)\\ m_{5}\left(1\right)=\;\mathrm{mod}\;\left(\left\lfloor \frac{\begin{array}{c} \left(a_{11}a_{22}+a_{12}a_{22}\right)c_{0}^{2}+(a_{11}a_{21}\varepsilon_{1}\\ +a_{12}a_{21}\varepsilon_{1}+a_{32})c_{0}+a_{31}\varepsilon_{2} \end{array}}{2^{16}}\right\rfloor ,2^{8}\right)\\ m_{6}\left(1\right)=\;\mathrm{mod}\;\left(\left\lfloor \frac{\begin{array}{c} \left(a_{11}a_{23}+a_{13}a_{22}\right)c_{0}^{2}+(a_{11}a_{21}\varepsilon_{1}\\ +a_{13}a_{21}\varepsilon_{1}+a_{33})c_{0}+a_{31}\varepsilon_{2} \end{array}}{2^{16}}\right\rfloor ,2^{8}\right)\\ m_{7}\left(1\right)=\;\mathrm{mod}\;\left(\left\lfloor \frac{\begin{array}{c} \left(a_{11}a_{24}+a_{14}a_{24}\right)c_{0}^{2}+(a_{11}a_{21}\varepsilon_{1}\\ +a_{14}a_{21}\varepsilon_{1}+a_{34})c_{0}+a_{31}\varepsilon_{2} \end{array}}{2^{16}}\right\rfloor ,2^{8}\right)\\ m_{8}\left(1\right)=\;\mathrm{mod}\;\left(\left\lfloor \frac{\begin{array}{c} \left(a_{12}a_{22}+a_{12}a_{23}+a_{13}a_{22}+a_{13}a_{23}\right)c_{0}^{2}+\\ \left(a_{12}a_{21}\varepsilon_{1}+a_{13}a_{21}\varepsilon_{1}+a_{32}+a_{33}\right)c_{0}+a_{31}\varepsilon_{2} \end{array}}{2^{16}}\right\rfloor ,2^{8}\right)\;\;\;\;\;\;\;\\ m_{9}\left(1\right)=\;\mathrm{mod}\;\left(\left\lfloor \frac{\begin{array}{c} \left(a_{12}a_{22}+a_{12}a_{24}+a_{14}a_{22}+a_{14}a_{24}\right)c_{0}^{2}+\\ \left(a_{12}a_{21}\varepsilon_{1}+a_{14}a_{21}\varepsilon_{1}+a_{32}+a_{34}\right)c_{0}+a_{31}\varepsilon_{2} \end{array}}{2^{16}}\right\rfloor ,2^{8}\right)\\ m_{10}\left(1\right)=\;\mathrm{mod}\;\left(\left\lfloor \frac{\begin{array}{c} \left(a_{13}a_{23}+a_{13}a_{24}+a_{14}a_{23}+a_{14}a_{24}\right)c_{0}^{2}+\\ \left(a_{13}a_{21}\varepsilon_{1}+a_{14}a_{21}\varepsilon_{1}+a_{33}+a_{34}\right)c_{0}+a_{31}\varepsilon_{2} \end{array}}{2^{16}}\right\rfloor ,2^{8}\right)\\ m_{11}\left(1\right)=\;\mathrm{mod}\;\left(\left\lfloor \frac{\begin{array}{c} \left(a_{11}a_{22}+a_{11}a_{23}+a_{12}a_{22}+a_{12}a_{23}+a_{13}a_{22}+a_{13}a_{23}\right)c_{0}^{2}+\\ \left(a_{11}a_{21}\varepsilon_{1}+a_{12}a_{21}\varepsilon_{1}+a_{13}a_{21}\varepsilon_{1}+a_{32}+a_{33}\right)c_{0}+a_{31}\varepsilon_{2} \end{array}}{2^{16}}\right\rfloor ,2^{8}\right)\\ m_{12}\left(1\right)=\;\mathrm{mod}\;\left(\left\lfloor \frac{\begin{array}{c} \left(a_{11}a_{22}+a_{11}a_{24}+a_{12}a_{22}+a_{12}a_{24}+a_{14}a_{22}+a_{14}a_{24}\right)c_{0}^{2}+\\ \left(a_{11}a_{21}\varepsilon_{1}+a_{12}a_{21}\varepsilon_{1}+a_{14}a_{21}\varepsilon_{1}+a_{32}+a_{34}\right)c_{0}+a_{31}\varepsilon_{2} \end{array}}{2^{16}}\right\rfloor ,2^{8}\right)\\ m_{13}\left(1\right)=\;\mathrm{mod}\;\left(\left\lfloor \frac{\begin{array}{c} \left(a_{11}a_{23}+a_{11}a_{24}+a_{13}a_{23}+a_{13}a_{24}+a_{14}a_{23}+a_{14}a_{24}\right)c_{0}^{2}+\\ \left(a_{11}a_{21}\varepsilon_{1}+a_{13}a_{21}\varepsilon_{1}+a_{14}a_{21}\varepsilon_{1}+a_{33}+a_{34}\right)c_{0}+a_{31}\varepsilon_{2} \end{array}}{2^{16}}\right\rfloor ,2^{8}\right)\\ m_{14}\left(1\right)=\;\mathrm{mod}\;\left(\left\lfloor \frac{\begin{array}{c} (a_{12}a_{22}+a_{12}a_{23}+a_{12}a_{24}+a_{13}a_{22}+a_{13}a_{23}+a_{13}a_{24}+a_{14}a_{22}+a_{14}a_{23}\\ +a_{14}a_{24})c_{0}^{2}+\left(a_{12}a_{21}\varepsilon_{1}+a_{13}a_{21}\varepsilon_{1}+a_{14}a_{21}\varepsilon_{1}+a_{32}+a_{33}+a_{34}\right)c_{0}+a_{31}\varepsilon_{2} \end{array}}{2^{16}}\right\rfloor ,2^{8}\right)\\ m_{15}\left(1\right)=\;\mathrm{mod}\;\left(\left\lfloor \frac{\begin{array}{c} (a_{11}a_{22}+a_{11}a_{23}+a_{11}a_{24}+a_{12}a_{22}+a_{12}a_{23}+a_{12}a_{24}+a_{13}a_{22}+a_{13}a_{23}+a_{13}a_{24}+a_{14}a_{22}\\ +a_{14}a_{23}+a_{14}a_{24})c_{0}^{2}+\left(a_{11}a_{21}\varepsilon_{1}+a_{12}a_{21}\varepsilon_{1}+a_{13}a_{21}\varepsilon_{1}+a_{14}a_{21}\varepsilon_{1}+a_{32}+a_{33}+a_{34}\right)c_{0}+a_{31}\varepsilon_{2} \end{array}}{2^{16}}\right\rfloor ,2^{8}\right) \end{array}\right.$$

From Equation (Equation 23), the second decryption results $m_{i}\left(1\right)\;\left(i=1,2,\cdots ,15\right)$ both contain an independent additive term $\left(a_{31}\varepsilon_{2}/2^{16}\right)$ that does not multiply with the initial condition under fifteen different initial conditions. When the initial conditions are set as a same non-zero constant $c_{0}$, the constant common factor multiplied by the secret key expression cannot be extracted from the decryption expression. Therefore, it is impossible to obtain the correct sub-block information of the secret key expression by selecting suitable values of initial conditions $c_{i}\left(i=1,2,3,4\right)$. Similarly, as the number of iterations *k* increases, $m_{i}\left(k\right)\;\left(k=2,3,\cdots \right)$ also contain the above independent addition term. Therefore, the method that combines the chosen-ciphertext attack and the divide-and-conquer attack proposed in [22,35] fails in this case.

In summary, the improved 4-D SCSCA-CAC proposed in this paper is safe against the combinational effect of the chosen-ciphertext attack and the divide-and-conquer attack.

## 3. Implementation of a Bitstream-Oriented Encrypted Video Communication System

Transmission frame rate is an important indicator to represent the real-time performance of video communication systems, and 25 f/s is a basic requirement in actual application scenarios. Thus, promoting real-time performance is a primary goal in system design. In this paper, an optimized design is proposed that can be conducted from three aspects: (1) improve the BOE method for further reducing the computational load; (2) utilize the hardware codec accelerator on the chip to speed the system; and (3) adopt the multi-core and multi-threading technology to improve system efficiency in parallel work.

### 3.1. Overall Design Scheme

The overall design scheme of the real-time video communication system based on the ARM platform and the BOE method is shown in Figure 2. On the sender, firstly, YUV raw videos are captured from the camera continuously and then encoded as the H.264 bitstream by hardware codec. Secondly, the H.264 bitstream is encrypted through the BOE method. Thirdly, the encrypted bitstream is sent to the receiver through Ethernet. On the receiver, the encrypted bitstream is received from the sender through Ethernet and then decrypted into the original H.264 bitstream through the bitstream-oriented decryption (BOD) method. Fourthly, the original bitstream is decoded into the original YUV videos by hardware codec. Finally, the original videos are displayed on the screen. When K is switched to channel 1, the encrypted bitstream can be decrypted correctly. When K is switched to channel 2, the encrypted bitstream is bypassed without decryption. Most significantly, the encrypted bitstream can be successfully decoded and displayed, which suggests that the encrypted bitstream keeps the H.264 format compatibility and can be transcoded by servers when it is transmitted through the Internet.

The BOE and BOD are specific implementations for selective encryption and decryption, respectively. There are three main parts in the BOE and BOD modules in this paper, including H.264 entropy decoding, chaos encryption-decryption, and H.264 entropy encoding. In the chaos encryption-decryption part, the proposed 4-D SCSCA-CAC is used to encrypt and decrypt the H.264 bitstream. According to Equation (Equation 16), the 4-D SCSCA-CAC expression for the BOE and BOD module are obtained as Equations (Equation 24) and (Equation 25), respectively, and a block diagram of 4-D SCSCA-CAC encryption-decryption in BOE and BOD modules is shown in Figure 2.
(24)$$\left\{\begin{array}{c} x_{1}^{e}\left(k+1\right)=a_{11}^{e}x_{1}^{e}\left(k\right)+a_{12}^{e}x_{{}_{2}}^{e}\left(k\right)+a_{13}^{e}x_{3}^{e}\left(k\right)+a_{14}^{e}x_{4}^{e}\left(k\right)\\ x_{2}^{e}\left(k+1\right)=a_{21}^{e}\varepsilon_{1}^{e}\cos\left(\sigma_{1}^{e}p\left(k\right)\right)+a_{22}^{e}x_{2}^{e}\left(k\right)+a_{23}^{e}x_{3}^{e}\left(k\right)+a_{24}^{e}x_{4}^{e}\left(k\right)\\ x_{3}^{e}\left(k+1\right)=a_{31}^{e}\varepsilon_{2}^{e}\cos\left(\sigma_{2}^{e}p\left(k\right)\right)+a_{32}^{e}x_{2}^{e}\left(k\right)+a_{33}^{e}x_{3}^{e}\left(k\right)+a_{34}^{e}x_{4}^{e}\left(k\right)\\ x_{4}^{e}\left(k+1\right)=a_{41}^{e}\varepsilon_{3}^{e}\cos\left(\sigma_{3}^{e}p\left(k\right)\right)+a_{42}^{e}x_{2}^{e}\left(k\right)+a_{43}^{e}x_{3}^{e}\left(k\right)+a_{44}^{e}x_{4}^{e}\left(k\right) \end{array}\right.$$
(25)$$\left\{\begin{array}{c} x_{1}^{d}\left(k+1\right)=a_{11}^{d}x_{1}^{d}\left(k\right)+a_{12}^{d}x_{{}_{2}}^{d}\left(k\right)+a_{13}^{d}x_{3}^{d}\left(k\right)+a_{14}^{d}x_{4}^{d}\left(k\right)\\ x_{2}^{d}\left(k+1\right)=a_{21}^{d}\varepsilon_{1}^{d}\cos\left(\sigma_{1}^{d}p\left(k\right)\right)+a_{22}^{d}x_{2}^{d}\left(k\right)+a_{23}^{d}x_{3}^{d}\left(k\right)+a_{24}^{d}x_{4}^{d}\left(k\right)\\ x_{3}^{d}\left(k+1\right)=a_{31}^{d}\varepsilon_{2}^{d}\cos\left(\sigma_{2}^{d}p\left(k\right)\right)+a_{32}^{d}x_{2}^{d}\left(k\right)+a_{33}^{d}x_{3}^{d}\left(k\right)+a_{34}^{d}x_{4}^{d}\left(k\right)\\ x_{4}^{d}\left(k+1\right)=a_{41}^{d}\varepsilon_{3}^{d}\cos\left(\sigma_{3}^{d}p\left(k\right)\right)+a_{42}^{d}x_{2}^{d}\left(k\right)+a_{43}^{d}x_{3}^{d}\left(k\right)+a_{44}^{d}x_{4}^{d}\left(k\right) \end{array}\right.$$
where $x_{i}^{e}\left(k\right)\;\left(i=1,2,3,4\right)$ represents the encryption chaotic variables, and $a_{ij}^{e}\;\left(1\leq i\leq 4,1\leq j\leq 4\right),\varepsilon_{i}^{e},\sigma_{i}^{e}\;\left(1\leq i\leq 3\right)$ denote the secret keys at encryption end. $x_{i}^{d}\left(k\right)\;\left(i=1,2,3,4\right)$ represents the decryption chaotic variables, and $a_{ij}^{d}\;\left(1\leq i\leq 4,1\leq j\leq 4\right),\varepsilon_{i}^{d},\sigma_{i}^{d}\;\left(1\leq i\leq 3\right)$ denote the secret keys at decryption end. Since 4-D SCSCA-CAC is a symmetric cipher, the encryption and decryption end secret keys need to be matched as $a_{ij}^{e}=a_{ij}^{d}=a_{ij}\;\left(1\leq i\leq 4,1\leq j\leq 4\right)$ and $\varepsilon_{i}^{e}=\varepsilon_{i}^{d}=\varepsilon_{i},\sigma_{i}^{e}=\sigma_{i}^{d}=\sigma_{i}\;\left(1\leq i\leq 3\right)$ to decrypt correctly.

On the sender, as shown in Figure 3, $\left\lfloor \cdot \right\rfloor$ represents round-down operation, ⊕ represents the bitwise XOR operation, $\;\mathrm{mod}\;(\cdot ,2^{8})$ is used for intercepting the lower eight-bits on $\left\lfloor \left(x_{1}^{\left(e\right)}\left(k\right)\times x_{2}^{\left(e\right)}\left(k\right)+x_{3}^{\left(e\right)}\left(k\right)\right)/2^{16}\right\rfloor$ to generate eight-bits encryption pseudo-random sequence $s^{e}\left(k\right)$. Then, the binary information of the original syntax element is encrypted as the encrypted H.264 syntax element p(k) by bitwise XOR with $s^{e}\left(k\right)$, hence the encryption operation is given by Equation (Equation 26). Finally, p(k) is fed back and substituted for $x_{1}^{e}\left(k\right)$ in the second to third equations of the chaotic system.
(26)$$p\left(k\right)=s^{e}\left(k\right)⊕m\left(k\right)=\;\mathrm{mod}\;\left(\left\lfloor \frac{x_{1}^{e}\left(k\right)\times x_{2}^{e}\left(k\right)+x_{3}^{e}\left(k\right)}{2^{16}}\right\rfloor ,2^{8}\right)⊕m\left(k\right)$$

On the receiver, the received p(k) is also substituted for $x_{1}^{d}\left(k\right)$ in the second to third equations of the chaotic system, $\;\mathrm{mod}\;(\cdot ,2^{8})$ is used for intercepting the lower eight-bits on $\left\lfloor \left(x_{1}^{\left(d\right)}\left(k\right)\times x_{2}^{\left(d\right)}\left(k\right)+x_{3}^{\left(d\right)}\left(k\right)\right)/2^{16}\right\rfloor$ to generate eight-bits decryption pseudo-random sequence $s^{d}\left(k\right)$. Similarly, the encrypted H.264 syntax element p(k) is decrypted as the original H.264 syntax element m(k) by bitwise XOR with $s^{d}\left(k\right)$. Therefore, the encryption operation is given by:(27)$$m\left(k\right)=s^{d}\left(k\right)⊕p\left(k\right)=\;\mathrm{mod}\;\left(\left\lfloor \frac{x_{1}^{d}\left(k\right)\times x_{2}^{d}\left(k\right)+x_{3}^{d}\left(k\right)}{2^{16}}\right\rfloor ,2^{8}\right)⊕p\left(k\right)$$

In Figure 2, the hardware codecs can be applied in the independent architecture separating apart the encryption algorithm and the video codec. The acceleration function of the hardware codec copes with the speed bottleneck caused by the software codec.

Multi-core multi-threading technology is another essential factor for improving real-time performance. The whole system can be divided into multiple threads. Each thread is executed by their according CPU in a parallel and pipeline manner. Better than the single thread operation, multi-core multi-threading technology is capable of solving the blocking and delay problems effectively and is beneficial to the execution efficiency.

### 3.2. Specific Design of Bitstream-Oriented Encryption Scheme for H.264/AVC

#### 3.2.1. Presentation of H.264/AVC Standard

Established by the Moving Picture Experts Group (MPEG) consortium and the Video Coding Expert Group (VCEG), H.264/AVC is one of the most popular standards for video encoding. With the characteristics of low bit rate and high compression rate, H.264/AVC is widely used in video applications. H.264/AVC-based standard specifies three profiles, including baseline profile, main profile, and extended profile, in which each profile supports a specific type of application. The baseline profile is mainly used for high real-time video wireless communication application scenarios, while the main profile and extended profile are used for applications with relatively low real-time requirements such as video broadcasting or video storage. Therefore, to satisfy the real-time performance of the video confidential communication system, one considers the baseline profile in this paper. In this profile, only inter prediction and intra prediction are supported, and the H.264 bitstream is encoded by the CALVC and Exponential-Golomb coding methods.

#### 3.2.2. Encryption Analysis of Syntax Elements in H.264 Bitstream

The H.264 bitstream is highly compressed bit sequences, and each bit data is closely related to the video encoding and decoding process. If arbitrary bit data is encrypted, the format compatibility of the encoded video may be destroyed. To achieve a format compatible and secure encryption scheme, firstly, one needs to understand the hierarchical structure of the H.264 bitstream and perform the encryption analysis for the syntax elements in H.264 bitstream. Finally, the syntax elements that can keep the video format compatibility after encryption will be selected as the encryption objects.

The hierarchical structure principle of the H.264 bitstream is shown in Figure 4. The H.264 bitstream is composed of multiple groups of pictures (GOP), and each GOP is composed of a series of frame pictures. In the baseline profile standard, the frame types in GOP only contain the I-frame and P-frame. An I-frame and P-frame are composed of one or more I-slices and P-slices, respectively. Each slice can be further divided into several macroblocks and one macroblock is composed of 16 × 16 pixel matrixes. Among them, an I-slice only contains I-macroblocks, while a P-slice contains both I-macroblocks and P-macroblocks. The I-macroblocks, intra prediction macroblocks, are encoded according to the decoded pixels in the current slice as the reference. The P-macroblocks, inter prediction macroblocks, are encoded according to the previous encoded picture as the reference.

In the H.264 bitstream, the syntax element is the basic unit of data, and each syntax element consisting of several bits represents a specific physical meaning. Different syntax elements are carried on the different types of the macroblock. For I-macroblock, there are Macroblock Type (Mb_type), Macroblock Prediction mode (Mb_pred), Coded Block Pattern (CBP), Quantization Parameter (QP), and Residual syntax elements. For P-macroblock, there are Mb_type, MVD, CBP, QP, and Residual syntax elements. The syntax element ‘Residual’, and other syntax elements, can be extracted by CAVLC entropy decoding and the Exponential-Golomb decoding method from the macroblock, respectively. Next, the above syntax elements will be analyzed one by one to determine whether they can be used for encryption.

• Mb_type specifies the type of the current macroblock. If the Mb_type is encrypted, the encrypted macroblock cannot be recognized by the codec, so the Mb_type syntax element cannot be encrypted.

• Mb_pred represents the prediction mode used to reconstruct the current block. Because the prediction modes applicable to macroblocks in different locations are different, if the encrypted value of Mb_pred is not within the applicable prediction mode for the current macroblock, it will also lead to decoding failure, so the Mb_pred syntax element cannot be encrypted.

• CBP refers to the coding scheme of the residual data of the current macroblock, each bit of which represents the number of the luma component or chroma component in the current macroblock. It is related to whether there is luma or chroma component data in the bitstream. Therefore, the CBP syntax element cannot be encrypted.

• QP is used to scale the prediction residual transform coefficients. Since the value of QP is limited in size, encrypting QP will cause greater data expansion. Therefore, the QP syntax element is not regarded as the encryption object in this paper.

• Residual carries the main data of the macroblock, which includes the DC component and the Alternating Current (AC) component. Among them, the DC component has most of the information of the video image. If the DC component is encrypted, the image information can be encrypted to the maximum extent. Thus, DC component in Residual syntax element is the main encryption object.

• MVD only exists in the P-macroblock, and it represents the motion direction of each sub-block in the current macroblock. Encrypting MVD will affect the reconstruction effect of the inter prediction image, so it can be encrypted.

#### 3.2.3. Design of BOE Module

The BOE module is made up of entropy decoding, chaotic encryption, and entropy encoding. The design flow of the module is as follows:


**Step1: Parse the syntax elements from original H.264 bitstream**


The macroblock $MB_{i}\;\left(1\leq i\leq n\right)$ as a basic processing unit in a frame of the H.264 video is parsed into the syntax elements. For I-macroblock, the ‘Residual’ syntax element is parsed through CAVLC entropy decoding for extracting the DC components $D_{j}^{I}\;\left(1\leq j\leq q\right)$ as the encryption objects, where $D_{j}^{I}$ represents the $j_{\mathrm{th}}$ DC component in a I-macroblock. The number of DC components *q* in a I-macroblock is determined by Coded Block Pattern Chroma (CBPC) and Coded Block Pattern Luma (CBPL). The values of CBPC and CBPL and the corresponding value of *q* have the following four situations: ① When CBPC=0 and CBPL=0, q=0. ② When CBPC=0 and CBPL=15, q=8. ③ When CBPC=1or2 and CBPL=0, q=16. ④ When CBPC=1or2 and CBPL=15, q=24.

For P-macroblock, DC components $D_{j}^{P}\;\left(1\leq j\leq q\right)$ and MVD coefficients $\mathrm{MV}D_{j}\;\left(1\leq j\leq z\right)$ as the encryption objects are extracted via CALVC and Exponential-Golomb entropy decoding operations, respectively. $D_{j}^{P}$ represents the $j_{\mathrm{th}}$ DC component in a P-macroblock. The same as the I-macroblock, the number of DC components in a P-macroblock is also determined by CBPC and CBPL. $\mathrm{MV}D_{j}$ represents the $j_{\mathrm{th}}$ MVD coefficient in a P-macroblock, which can be further divided into horizontal component $Mx_{j}$ and vertical component $My_{j}$. The quantity of the MVD coefficients represented as *z* is determined by the division mode of the P-macroblock. When the division mode is ① P_16×16, z=1. ② When P_16×8 or P_8×16, z=2. ③ When P_8×8, z=4. ③ When P_4×8 or P_8×4, z=8. ④ When P_4×4, z=16. Notably, the division mode is decided by the Mb_type syntax element.


**Step2: Encrypt the syntax elements**


The syntax elements are encrypted by 4-D SCSCA-CAC. When the current frame is an I-frame, the DC component in the I-macroblock is the only encrypted object. According to Equation (Equation 26), the encryption expression is as follows:(28)$$p\left(k\right)=s^{e}\left(k\right)⊕m\left(k\right)=s^{e}\left(k\right)⊕D_{ij}^{I}\left(k\right)={\hat{D}}_{ij}^{I}\left(k\right)$$
where $s^{e}\left(k\right)$ represents the encryption sequence which is expressed as $s^{e}\left(k\right)=\;\mathrm{mod}\;\left(\left\lfloor \left(x_{1}^{e}\left(k\right)\times x_{2}^{e}\left(k\right)+x_{3}^{e}\left(k\right)\right)/2^{16}\right\rfloor ,2^{8}\right)$, $D_{ij}^{I}\;\left(1\leq i\leq n,1\leq j\leq q\right)$ represents the $j_{th}$ DC component in the $i_{th}$ I-macroblock in a I-frame, and ${\hat{D}}_{ij}^{I}$ represents the encrypted DC component corresponding to $D_{ij}^{I}$. Equation (Equation 28) requires $k=\sum_{i=1}^{n}q_{i}$ iterations to fulfill encryption operation for one I-frame, where $q_{i}$ represents the quantity of the DC components in the current I-macroblock.

When the current frame is a P-frame, there are two types of macroblocks to be considered. In the I-macroblock, only the DC components are encrypted. In the P-macroblock, both the DC components and MVD coefficients are encrypted. Assuming that the amount of the macroblocks in a P-frame is *n*, and the quantity of the P-macroblocks is m(m≤n) in the current frame, then the number of the I-macroblocks is n−m. The encryption expression is as follows: (29)$$p\left(k\right)=s^{e}\left(k\right)⊕m\left(k\right)=\left\{\begin{array}{c} s^{e}\left(k\right)⊕D_{pj}^{I}\left(k\right)={\hat{D}}_{pj}^{I}\left(k\right)\mathrm{when}\;\mathrm{MB}=I\\ s^{e}\left(k\right)⊕D_{pj}^{P}\left(k\right)={\hat{D}}_{pj}^{P}\left(k\right)\mathrm{when}\;\mathrm{MB}=P\\ s^{e}\left(k\right)⊕Mx_{pj}\left(k\right)=\hat{M}x_{pj}\left(k\right)\mathrm{when}\;\mathrm{MB}=P\\ s^{e}\left(k\right)⊕My_{pj}\left(k\right)=\hat{M}y_{pj}\left(k\right)\mathrm{when}\;\mathrm{MB}=P \end{array}\right.$$
where $D_{pj}^{I}\;\left(1\leq p\leq n-m,1\leq j\leq q\right)$ represents the $j_{th}$ DC component in the $p_{th}$ I-macroblock, and $D_{pj}^{P}\;\left(1\leq p\leq n,1\leq j\leq q\right)$ represents the $j_{th}$ DC component in the $p_{th}$ P-macroblock in a P-frame. $Mx_{pj}$ and $My_{pj}$ denote the horizontal component and vertical component of the $j_{th}$ MVD coefficient in the $p_{th}$ P-macroblock, and the corresponding ciphertexts are represented as $\hat{M}x_{pj}$ and $\hat{M}y_{pj}$, respectively. Equation (Equation 29) requires $k=\sum_{i=1}^{n}q_{i}+\sum_{i=1}^{m}2z_{i}$ iterations to complete encryption operation for one P-frame, where $z_{i}$ represents the quantity of the MVD coefficients in the current P-macroblock, and $q_{i}$ represents the quantity of the DC components in the current macroblock.


**Step3: Re-encode the encrypted syntax elements as the encrypted H.264 bitstream**


After encrypting, ${\hat{D}}_{ij}^{I}$, ${\hat{D}}_{pj}^{I}$, and ${\hat{D}}_{pj}^{P}$ are re-encoded via CAVLC entropy encoding operation. Meanwhile, $\hat{M}x_{pj}$ and $\hat{M}y_{pj}$ are re-encoded through the Exponential-Golomb encoding operation. Finally, the encrypted bitstream is substituted for the original H.264 bitstream. Remarkably, excepting for the syntax elements of the DC components and MVD coefficients, the remaining syntax elements remain unchanged during the above encryption process.

From the above description, one knows that encryption iteration *k* in the BOE method is much less than the FE method. Hence, the BOE method with advantages of lower calculation load and higher processing speed obviously enhances the real-time performance of the system. The execution flow of the BOE module is shown in Algorithm 1.
**Algorithm 1:** BOE operating procedures
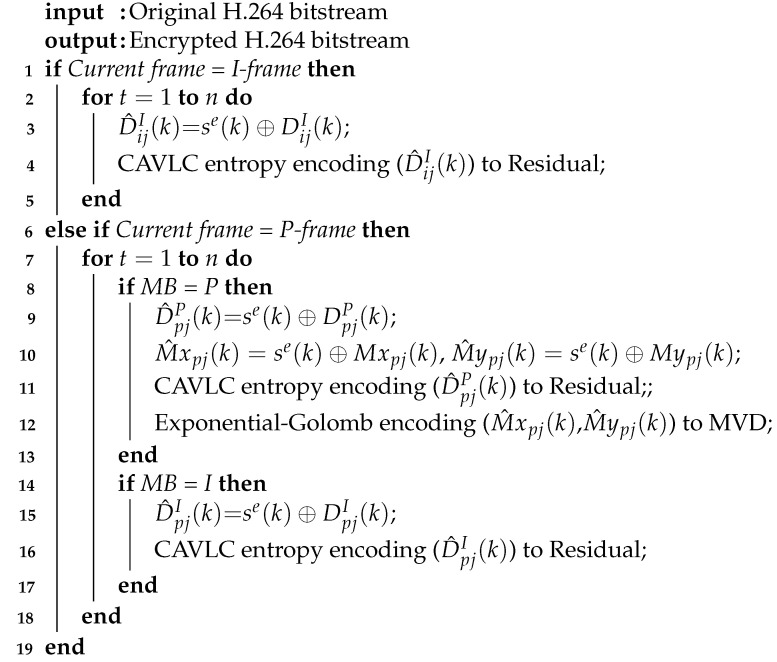


#### 3.2.4. Design of BOD Module

The BOD module is made up of entropy decoding, chaotic decryption, and entropy encoding. The design flow of the module is as follows:


**Step1: Parse the syntax elements from encrypted H.264 bitstream**


Parse the encrypted syntax elements from macroblock $MB_{i}\;\left(1\leq i\leq n\right)$ in a frame of the encrypted H.264 video. For I-macroblocks, ${\hat{D}}_{j}^{I}\;\left(1\leq j\leq q\right)$ are extracted from the ‘Residual’ syntax element. For P-microblock, in addition to obtaining ${\hat{D}}_{j}^{P}\;\left(1\leq j\leq q\right)$, it still has to extract $\mathrm{MV}D_{j}\;\left(1\leq j\leq z\right)$.


**Step2: Decrypt the encrypted syntax elements**


The encrypted syntax elements are also decrypted by 4-D SCSCA-CAC. When the current frame is an I-frame, the decryption expression according to Equation (Equation 27) is as follows:(30)$$D_{ij}^{I}\left(k\right)=s^{d}\left(k\right)⊕p\left(k\right)=s^{d}\left(k\right)⊕{\hat{D}}_{ij}^{I}\left(k\right)$$
where s(k) represents the decryption sequence which is expressed as $s^{d}\left(k\right)=\;\mathrm{mod}\;\left(\left\lfloor \left(x_{1}^{d}\left(k\right)\times x_{2}^{d}\left(k\right)+x_{3}^{d}\left(k\right)\right)/2^{16}\right\rfloor ,2^{8}\right)$. Equation (Equation 30) requires $k=\sum_{i=1}^{n}q_{i}$ iterations to complete the decryption operation for one I-frame.

When the current frame is a P-frame, there are two types of macroblocks to be considered. In the I-macroblock, only the encrypted DC components need to be decrypted. In the P-macroblock, both the encrypted DC components and MVD coefficients need to be decrypted. The decryption expression is as follows:(31)$$m\left(k\right)=\left\{\begin{array}{c} s^{d}\left(k\right)⊕p\left(k\right)=s^{d}\left(k\right)⊕{\hat{D}}_{pj}^{I}\left(k\right)\begin{array}{c} =D_{pj}^{I}\left(k\right) \end{array}\mathrm{when}\;\mathrm{MB}\begin{array}{c} = \end{array}I\\ s^{d}\left(k\right)⊕p\left(k\right)=s^{d}\left(k\right)⊕{\hat{D}}_{pj}^{P}\left(k\right)\begin{array}{c} =D_{pj}^{P}\left(k\right) \end{array}\mathrm{when}\;\mathrm{MB}\begin{array}{c} = \end{array}P\\ s^{d}\left(k\right)⊕p\left(k\right)=s^{d}\left(k\right)⊕\hat{M}x_{pj}\left(k\right)\begin{array}{c} =Mx_{pj}\left(k\right) \end{array}\mathrm{when}\;\mathrm{MB}\begin{array}{c} = \end{array}P\\ s^{d}\left(k\right)⊕p\left(k\right)=s^{d}\left(k\right)⊕\hat{M}y_{pj}\left(k\right)\begin{array}{c} =My_{pj}\left(k\right) \end{array}\mathrm{when}\;\mathrm{MB}\begin{array}{c} = \end{array}P \end{array}\right.$$


**Step3: Re-encode the syntax elements as the original H.264 bitstream**


After decryption, $D_{ij}^{I}$, $D_{pj}^{I}$, and $D_{pj}^{P}$ are re-encoded via CALVC entropy encoding operation. Meanwhile, $Mx_{pj}$ and $My_{pj}$ are re-encoded through the Exponential-Golomb decoding operation. Last, the decrypted bitstream is substituted for the encrypted bitstream.

The execution flow of the BOD module is shown in Algorithm 2. Note that both the CAVLC entropy and the Exponential-Golomb coding operation in Algorithm 1 and Algorithm 2 are transplanted from JM86 and X264 software codec models.
**Algorithm 2:** BOD operating procedures
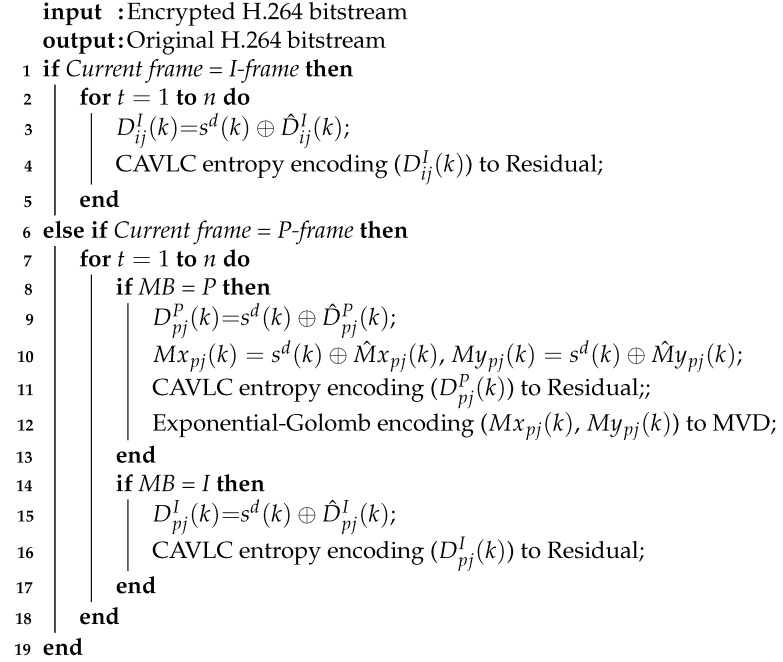


### 3.3. Multi-Core Multi-Threading Process

Multi-core multi-threading technology is an effective way to optimize system architecture and improve the efficiency of multitasking. The processing time for one frame on the sender includes video capturing $t_{cp}$, video encoding $t_{enc}$, encryption $t_{enp}$, video conversing $t_{cnv}$, and sending $t_{s}$. The hardware platform that is used in our implementation has four ARM Cortex A9 cores. In the case of single thread technology, the total operation time $T=t_{cp}+t_{enc}+t_{enp}+t_{cnv}+t_{s}$ is much longer than 40 ms. Obviously, the system cannot achieve the real-time metrics (25 f/s). In contrast, multi-core multi-threading technology splits the operation tasks into six threads. Under the reasonable assignment of the tasks, operation time for each thread $t_{i}\;\left(i=2,3\cdots \right)$ can be less than 40 ms. Therefore, the total operation time for one frame can fulfill T<40ms, reaching the basic frame rate indicator. The multi-core multi-threading design principle is shown in Figure 5. Figure 5a,b represents the multi-core multi-threading design schemes for the sender and receiver in Figure 2.

In Figure 5, different threads are assigned to the appropriate CPU for execution according to the computational complexity of each thread. Taking the sender as an example, Thread_*i* (*i* = 1,4,6) with low computational complexity are both performed by CPU1, while Thread_*i* (*i* = 2,3,5) with high computational complexity are executed by CPU2, CPU3, and CPU4, respectively. Moreover, threads exchange data through shared memories. Thread_*i* (*i* = 1,2,3) share Buffer_1, Thread_*i* (*i* = 2,3,4) share Buffer_2, Thread_*i* (*i* = 1,5) share Buffer_3, and Thread_*i* (*i* = 5,6) share Buffer_4. The buffer capacity is configured according to one frame size in YUV, RGB, or H.264 format.

Read–write conflict prevention is the most crucial working mechanism for shared buffers. It prohibits more than one thread to perform a reading or writing operation in the same buffer simultaneously. Otherwise, unpredictable errors will happen. The mutex lock mechanism can achieve the read–write conflict protection, whose working principle is illustrated by switches $K_{i}\;\left(i=1,2,\cdots 11\right)$ in Figure 4. Taking Buffer_1 in Figure 4a as an example, once $K_{1}$ is closed, $K_{2}$ and $K_{3}$ are kept open. It means that when Thread_1 is executing a data storage operation in Buffer_1, Thread_2 and Thread_3 are prohibited to operate Buffer_1 and carry out data format conversion for the previous frame at the same time.

In summary, the BOE and BOD modules have the advantages of high efficiency and good real-time performance. Besides, the hardware codec solves the time-consuming problem of encoding operation, and the multi-core multi-threading technology further optimizes the system. As a result, the operation period of all threads are constrained to less than 40 ms. Experiment results prove that the frame rate of the embedded hardware system is more than 25 f/s.

## 4. Experimental Results and Analysis

According to the chaotic encryption and decryption algorithms given in Equations (Equation 24) and (Equation 25), as well as the design principles of Figure 2, Figure 3, Figure 4 and Figure 5, one has realized the remote real-time video confidential communication system, as shown in Figure 6. First, choose two demo boards with four ARM Cortex A9 cores as the sender and receiver. Second, attach the camera and screens for video capturing and displaying. Third, connect two boards to LAN and set the IP address in the range of 192.168.1.1 to 192.168.1.255.

### 4.1. Experiment Results

The real-time video confidential communication system is tested in the real network environment under the 640×480 video resolution.

The experimental result of the original videos displaying at the sender is shown in Figure 7a. When keys are matched as $a_{ij}^{e}=a_{ij}^{d}=a_{ij}\;\left(1\leq i\leq 4,1\leq j\leq 4\right)$ and $\varepsilon_{i}^{e}=\varepsilon_{i}^{d}=\varepsilon_{i},\sigma_{i}^{e}=\sigma_{i}^{d}=\sigma_{i}\;\left(1\leq i\leq 3\right)$, the experimental result of successful decryption is shown in Figure 7b. When keys are mismatched, the experimental result of decryption failure is shown in Figure 7c. When encrypted videos are decoded and displayed without decryption, the experimental result is shown in Figure 7d. The experimental results show that the BOE method with the 4-D SCSCA-CAC can achieve effective perceptual encryption. Besides, the encrypted bitstream is available to be decoded without decryption. According to the test result, the average transmission frame rate of the system is up to 27 f/s.

The comparison of experiment results of the video confidential communication system based on ARM-embedded platform is shown in Table 3. The hardware platform for experiments in Table 3 is the ARM Cortex A9 core development board and the video resolution is 640 × 480. From the experimental results of [32,36], it can be seen that using the hardware codec and multi-core and multi-threaded technology can effectively improve the video transmission frame rate of the system. With the same multi-core and multi-threaded technology, although the FE scheme [32] can utilize the hardware codec to improve the video transmission frame rate, it cannot meet the video format compatibility. On the contrary, although the CIE scheme [36] can meet the video format compatibility, the transmission frame rate of this scheme is lower than 17 f/s caused by the software codec with a large computing load and time consumption. However, in the case of having two more operations of entropy coding and entropy decoding than the above schemes, the video transmission frame rate of the proposed BOE scheme not only reaches 27 f/s higher than that of the full encryption scheme, but also meets the format compatibility of video at the same time. Therefore, one can know that the proposed BOE scheme is superior in terms of security, real-time performance, and format compatibility.

### 4.2. Security Performance Tests

#### 4.2.1. NIST Test

NIST test is a strict and standardized test suite for testing the statistical characteristics of pseudo-random sequences generated by chaotic systems, and the total length of tested sequecnces are required to be at least $10^{8}$ bits. The NIST test includes 15 different tests, and the result of each test is represented by a *p*-value. Sequences are said to pass a test if the calculated *p*-value > 0.01.

Table 4 shows the result of the passing ratio and means of *p*-value of each test for 4-D SCSCA-CAC. In this experiment, the 100 groups of bit sequence with length as $10^{6}$ are generated from from intercepting the low 8-bit of the chaotic variable expressed as $s\left(k\right)\;=\;\mathrm{mod}\;\left(\left\lfloor \left(x_{1}\left(k\right)\times x_{2}\left(k\right)+x_{3}\left(k\right)\right)/2^{16}\right\rfloor ,2^{8}\right)$. From Table 4, one can have that the 4-D SCSCA-CAC can pass all the tests, which implies that the sequences generated from 4-D SCSCA-CAC are with good statistical performances and can be regarded as true random.

#### 4.2.2. TESTU01 Test

Compared with the NIST test, the TESTU01 test is a more rigorous statistical characteristic test. TESTU01 has seven test suites, including SmallCrush, Crush, BigCrush, Alphabit, Rabbit, PseudoDIEHARD, and FIPS-140-2 suite. Among them, the amount of test sequences of BigCrush suite is as high as 10TB, which makes the dynamic behavior of chaotic system easier to be exposed. Even though some chaotic systems can pass NIST test, they cannot further pass the TESTU01 test. Therefore, if the chaotic sequences generated by the chaotic system can successfully pass the TESTU01 test, it shows that the chaotic sequence has better statistical performance and randomness.

Table 5 shows the results of TESTU01 test for the 4-D SCSCA-CAC. The 10Tb test sequences are generated by intercepting the low 8-bit of the chaotic variable expressed as $s\left(k\right)\;=\;\mathrm{mod}\;\left(\left\lfloor \left(x_{1}\left(k\right)\times x_{2}\left(k\right)+x_{3}\left(k\right)\right)/2^{16}\right\rfloor ,2^{8}\right)$. The results show that the 4-D SCSCA-CAC can successfully pass all the tests in TESTU01.

#### 4.2.3. Phase Space Reconstruction Attacks

The phase space reconstruction attack is to analyze the time series of a state variable at different times, continuously produced by the chaotic system. It reconstructs the time series of the state variable by determining the appropriate delay time τ and embedding dimension *m*, and then recovers the regular trajectories such as attractors in the embedded dimension space. The structure of some chaotic maps with complicated trajectory will become simple and evident in the reconstructed phase space. Therefore, the attacker can easily predict the behavior of the chaotic variable according to the trajectory in the reconstructed phase space. Next, the phase space reconstruction attack is used to test the state variable $\left\{x_{1}\left(k\right)\right\}$ in 4-D SCSCA-CAC.

By using the auto-correlation method and False Nearest Neighbor (FNN) method, the delay time τ is calculated as 1 and the embedding dimension *m* is calculated as 4, and these two parameters are used to reconstruct the phase space. The estimating results of the delay time τ and the embedding dimension *m* are shown in Figure 8a,b, respectively. The reconstructed phase space is shown in Figure 9. It can be seen from Figure 9 that the reconstructed phase space is disordered and has no obvious structure, so the attacker cannot predict the behavior of the chaotic variable by reconstructing the phase space. Therefore, 4-D SCSCA-CAC can effectively resist the phase space reconstruction attack.

#### 4.2.4. Sensitivity Test of Key Parameters Mismatch

When the chaotic system has a key parameter with a very small mismatch error, and the original video signal is not able to decrypt, the key parameters are very sensitive to the mismatch error. The smaller the number of the mismatch error, the better the security of the system.

The sensitivity test results of key parameters mismatch for 4-D SCSCA-CAC is shown in Table 6. The absolute value of the mismatch between the key parameters in encryption system and decryption system is expressed as $|\Delta a_{ij}\left|=|a_{ij}^{e}-a_{ij}^{d}\right|\;\left(1\leq i\leq 4,1\leq j\leq 4\right)$. From the test results, one knows that any key parameter in 4-D SCSCA-CAC has high sensitivity to the tiny mismatch error. Therefore, 4-D SCSCA-CAC can effectively resist the brute force attacks.

#### 4.2.5. PSNR and SSIM Index Tests

PSNR can be used as a performance index to evaluate the perceptual security of encrypted video. When the PSNR value is lower than 20 dB, it means that original video information cannot be discriminated from the encrypted video by human eyes. PSNR is calculated based on the Mean Square Error (MSE) between the original image and the encrypted image. The mathematical expression of MSE is as follows:(32)$$\mathrm{MSE}=\frac{1}{H\times W}\sum_{x=0}^{H-1}\sum_{y=0}^{W-1}{\left(P_{org}\left(x,y\right)-P_{enc}\left(x,y\right)\right)}^{2}$$

The mathematical expression of PSNR is obtained as:(33)$$\mathrm{PSNR}=10\mathrm{lg}\left(\frac{{\left(2^{k}-1\right)}^{2}}{\mathrm{MSE}}\right)$$

Among them, *H* and *W* denote the height and width of the video, respectively. $P_{org}\left(x,y\right)$ represents the pixel value of the original video image, $P_{enc}\left(x,y\right)$ represents the pixel value of the encrypted video, and *k* represents the pixel depth.

Different from PSNR that measures image quality based on pixel error, SSIM measures image similarity from three aspects: brightness, contrast, and structure. The SSIM value is in the range of [0,1]. The smaller the SSIM value is, the lower the structural similarity will be, leading to higher perceptual security for the encrypted video.

In order to test the encryption effect of the video confidential communication system, one intercepted three frames of original images and their corresponding encrypted images in the actual video communication process, as shown in Figure 10. The PSNR and SSIM indicators test results are shown in Table 7.

According to Table 7, the PSNR values are much smaller than 20 dB, indicating that it is difficult to obtain the original frames’ information from the encrypted frames. The SSIM values are all less than 0.05, representing the low structural similarity between the original and encrypted images. All the above statistical analysis results come to the conclusion that the BOE scheme has good perceptual security performance.

## 5. Conclusions

For ensuring security, format compliance, and real-time transmission of encrypted videos, the SE method has become a research hotspot in the field of video encryption. In SE methods, with high computational efficiency, BOE is a more desirable encryption method compared with the CIE scheme. Some reports have studied and improved the BOE scheme, but these schemes lack the corresponding hardware implementation to prove the feasibility, effectiveness, and superiority of the BOE method. Moreover, some studies adopted AES block cipher as the encrypted algorithm in their BOE scheme with high computational load, increasing the time consumption in encrypting. To deal with these problems, in this paper, one proposed:

(1) An improved algorithm 4-D SCSCA-CAC.

(2) An improved BOE scheme utilizing the hardware codec to improve the real-time performance of video transmission.

(3) An ARM-based hardware implementation of the BOE scheme.

4-D SCSCA-CAC can resist the cryptanalysis combining of the chosen-ciphertext attack and the divide-and-conquer attack, and the chaotic bit sequences generated by 4-D SCSCA-CAC for encrypting video information have passed the NIST and TESTU01 test, ensuring the security of the video confidential communication system. In addition, due to the chaotic stream cipher having less computational overhead, without the consuming time for package construction to encrypt the fixed-length syntax elements, 4-D SCSCA-CAC is more appropriate for application in actual video communication than AES algorithm. Besides, one found that the hardware codec can be used for the BOE scheme in actual applications. The proposed scheme utilizes the hardware acceleration to improve the video transmission frame rate to 27 f/s. In contrast with hardware implementation based on the CIE method, the experimental results have proved that the BOE scheme is more suitable for real-time video secure communication application scenarios, and reflected the advantages of high efficiency and flexibility in the BOE scheme.

## Figures and Tables

**Figure 1 entropy-23-01431-f001:**
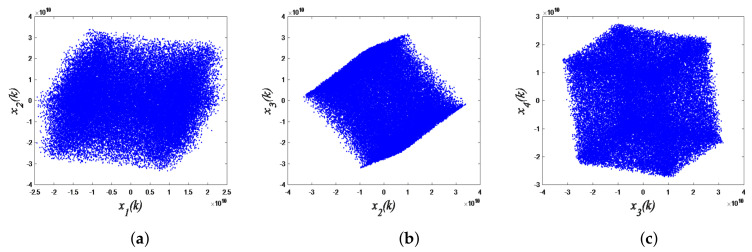
Phase diagram of chaos attractor: (**a**)$x_{1}\left(k\right)$ - $x_{2}\left(k\right)$ plane, (**b**) $x_{2}\left(k\right)$ - $x_{3}\left(k\right)$ plane, (**c**) $x_{3}\left(k\right)$ - $x_{4}\left(k\right)$ plane.

**Figure 2 entropy-23-01431-f002:**
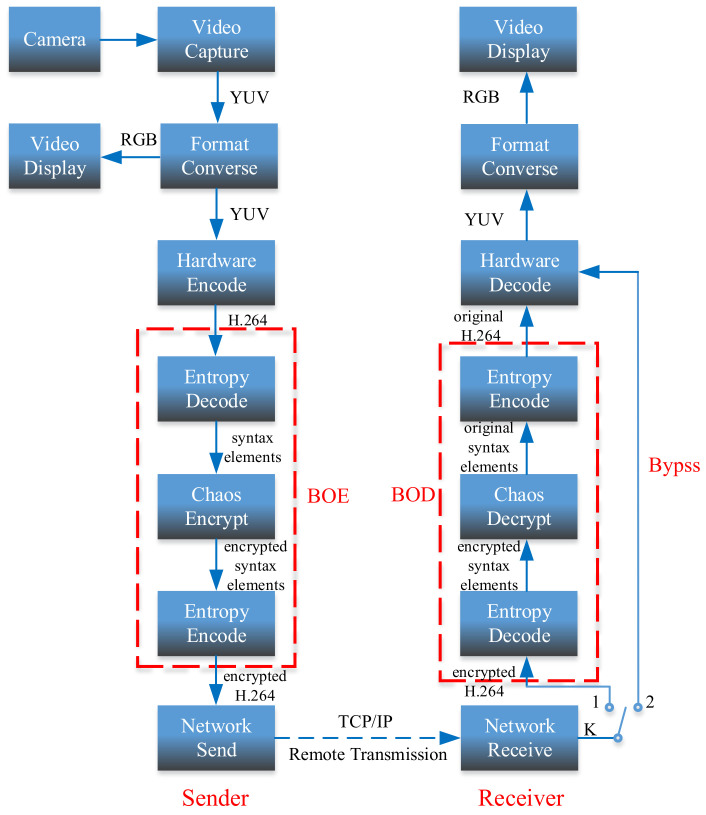
Block diagram of the hardware system design principle.

**Figure 3 entropy-23-01431-f003:**
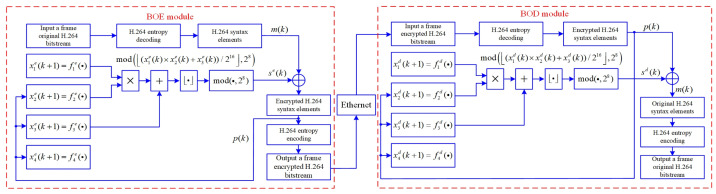
Block diagram of 4-D SCSCA-CAC encryption-decryption in BOE and BOD modules.

**Figure 4 entropy-23-01431-f004:**
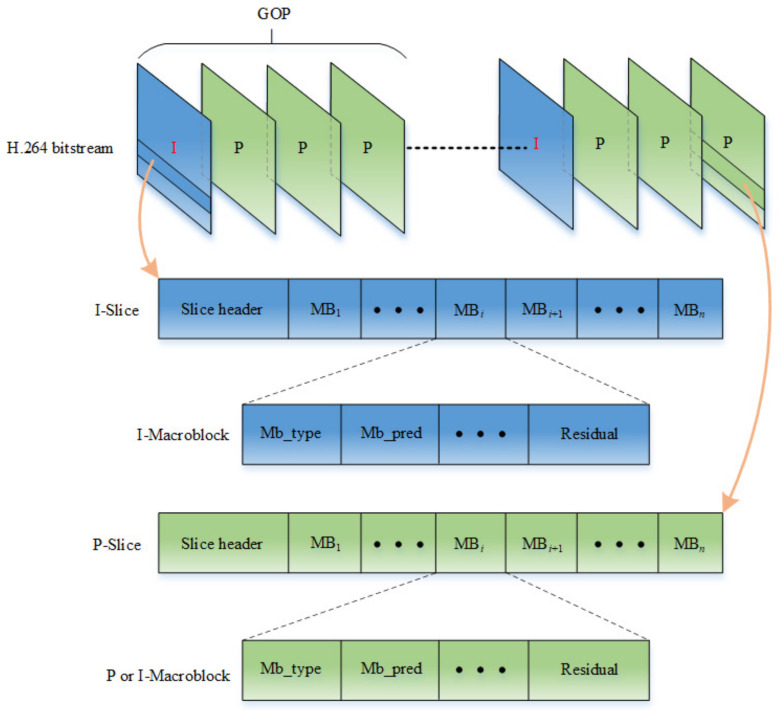
H.264 bitstream hierarchy principle.

**Figure 5 entropy-23-01431-f005:**
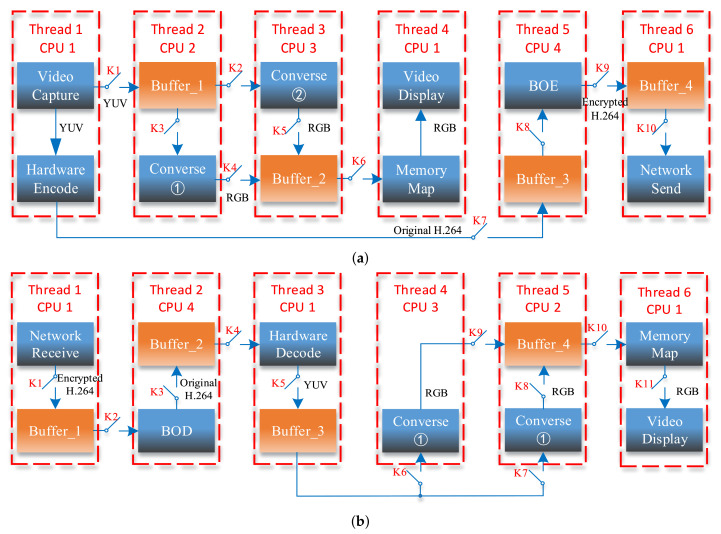
Multi-core multi-threading design principle: (**a**) sender, (**b**) receiver.

**Figure 6 entropy-23-01431-f006:**
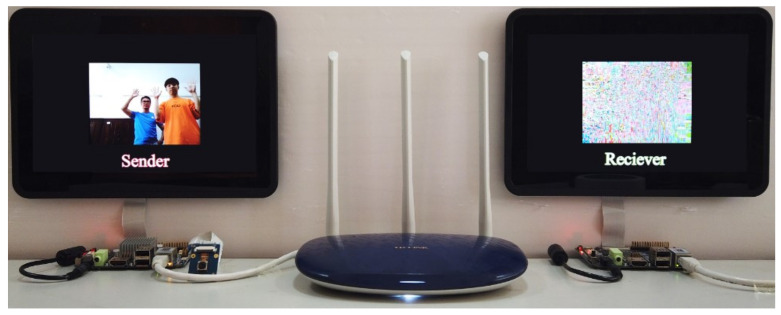
ARM-embedded system hardware platform.

**Figure 7 entropy-23-01431-f007:**
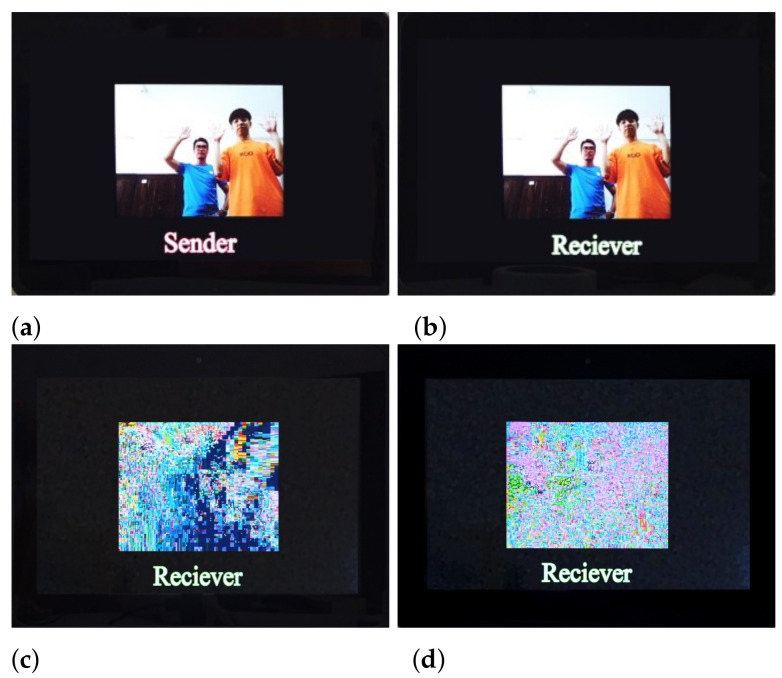
Experimental results of ARM-based embedded remote video confidential communication system: (**a**) original video image, (**b**) decrypted video when key matching, (**c**) decrypted video when key mismatching, (**d**) decoded video without decryption.

**Figure 8 entropy-23-01431-f008:**
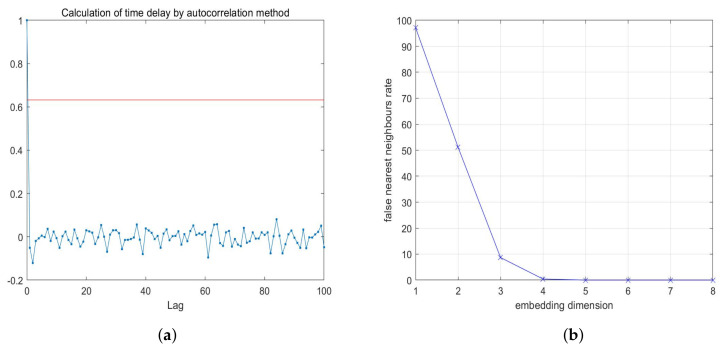
Estimating the delay time and the embedding dimension: (**a**) auto-correlation method for estimating τ, (**b**) FNN method for estimating *m*.

**Figure 9 entropy-23-01431-f009:**
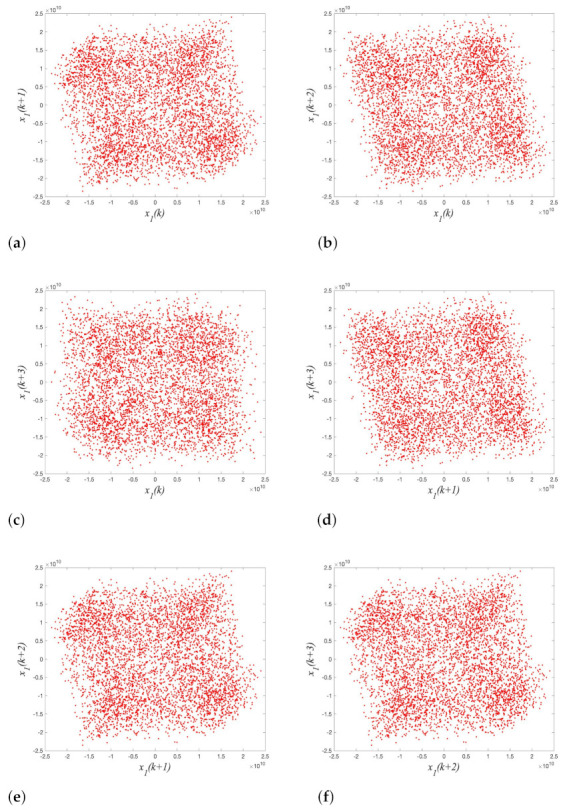
Phase space reconstruction for 4-D SCSCA-CAC: (**a**) $x_{1}\left(k\right)-x_{1}\left(k+1\right)$, (**b**) $x_{1}\left(k\right)-x_{1}\left(k+2\right)$, (**c**) $x_{1}\left(k\right)-x_{1}\left(k+3\right)$, (**d**) $x_{1}\left(k+1\right)-x_{1}\left(k+2\right)$, (**e**) $x_{1}\left(k+1\right)-x_{1}\left(k+3\right)$, (**f**) $x_{1}\left(k+2\right)-x_{1}\left(k+3\right)$.

**Figure 10 entropy-23-01431-f010:**
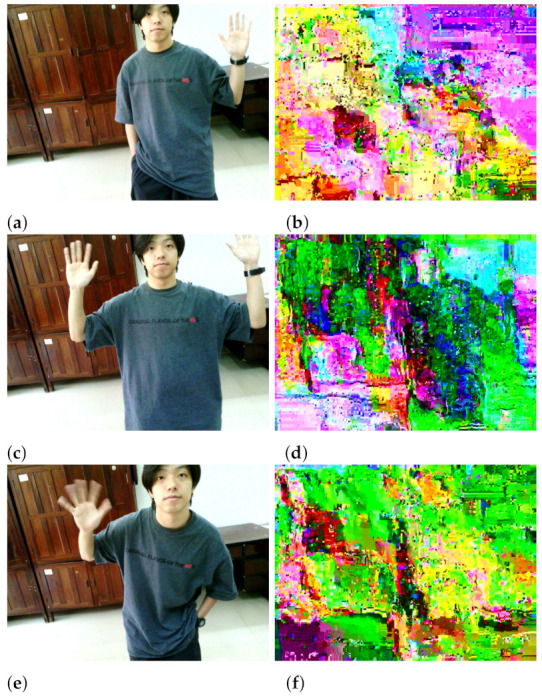
The original image and its corresponding encrypted image: (**a**) P1, (**b**) E1, (**c**) P2, (**d**) E2, (**e**) P3, (**f**) E3.

**Table 1 entropy-23-01431-t001:** Comparisons of the features of BOE, CIE, and FE methods.

Encryption Scheme	Encryption Stage	Input Data	Codec Form	Hardware Codec	Format Compatibility
BOE	After encoding	H.264 bitstream	Standardization	Available	Yes
CIE	During encoding	YUV raw video	Customization	Unavailable	Yes
FE	After encoding	H.264 bitstream	Standardization	Available	No

**Table 2 entropy-23-01431-t002:** The algorithm structure of *n*-D SCSCA (*n* = 3, 7, 8) and corresponding cryptanalysis methods.

Encryption Algorithm	Anti-Controller Type	Decryption Expression	Cryptanalysis Method	Analysis Results
8-D SCSCA [19]	$\begin{array}{c} \;\mathrm{mod}\;(\sigma_{i}p\left(k\right),\varepsilon_{i})\\ \;\;\;\;\;(i=1,2,3) \end{array}$	$\begin{array}{c} m(k)=s(k)⊕p(k)\\ =\;\mathrm{mod}\;\left(\left\lfloor x_{i}\left(k\right)\right\rfloor ,2^{8}\right)⊕p\left(k\right)\;\left(i=1,2,3\right) \end{array}$	Combination of chosen- ciphertext attack and DCA-TSNCIC [22]	Except the secret keys multiplied with the ciphertext and anti- controller secret keys, the rest of the secret keys in the encryption algorithms are deciphered.
3-D SCSCA [20]	εsin(σp(k))	$\begin{array}{c} m(k)=s(k)⊕p(k)\\ =\;\mathrm{mod}\;\left(\left\lfloor x_{1}\left(k\right)x_{2}\left(k\right)/2^{27}\right\rfloor ,2^{8}\right)⊕p\left(k\right) \end{array}$	Combination of chosen- ciphertext attack and DCA-TSNCIC [22]	
7-D SCSCA [21]	$\begin{array}{c} \;\mathrm{mod}\;(\sigma_{i}p\left(k\right),\varepsilon_{i})\\ \;\;(i=1,2,3) \end{array}$	$\begin{array}{c} m(k)=s(k)⊕p(k)\\ =\;\mathrm{mod}\;\left(\left\lfloor x_{i}\left(k\right)x_{j}\left(k\right)x_{l}\left(k\right)/2^{24}\right\rfloor ,2^{8}\right)⊕p\left(k\right)\\ \;(i=1,2,3;j=3,4,5;l=5,6,7) \end{array}$	Combination of chosen- ciphertext attack and DCA-TMNCIC [35]	

**Table 3 entropy-23-01431-t003:** Comparison of experiment results of video confidential communication system based on ARM-embedded platform.

Literature	Encryption Method	Format Compatibility	Video Revolution	Multi-Thread Operation	H.264 Codec	Frame Rate
Ours	Bitstream- oriented encryption	Yes	640×480	Six-thread	Hardware	27 f/s
[32]	Compression- integrated encryption	Yes	640×480	Four-thread	Software	17 f/s
[36]				One-thread	Software	2.51 f/s
[32]	Full encryption	No	640×480	Four-thread	Hardware	26.8 f/s
				One-thread	Hardware	14.68 f/s
				One-thread	Software	3.74 f/s

**Table 4 entropy-23-01431-t004:** NIST test results.

Test Index	Passing Ratio	Means of *p*-Value	Test Results
Frequency	0.96	0.137282	✓
Block frequency	1.00	0.678686	✓
Cumulative sums	0.97	0.425306	✓
Runs	1.00	0.236810	✓
Longest runs of ones	0.99	0.181557	✓
Rank	0.99	0.935716	✓
FFT	1.00	0.759756	✓
Non-overlapping templates	0.99	0.574903	✓
Overlapping templates	0.99	0.042808	✓
Universal	0.97	0.816537	✓
Approximate entropy	0.99	0.955835	✓
Random excursions	0.98	0.407091	✓
Random excursions variant	0.99	0.253551	✓
Serial	0.99	0.616264	✓
Linear complexity	1.00	0.514124	✓

**Table 5 entropy-23-01431-t005:** TESTU01 test results.

Test Suites	Data Size	Number of Tests	Test Results
SmallCrush	6 GB	15	✓
Crush	6 GB	144	✓
BigCrush	10 TB	160	✓
Alphabit	953 Mb	17	✓
Rabbit	953 Mb	40	✓
PseudoDIEHARD	6 GB	126	✓
FIPS-140-2	19 KB	16	✓

**Table 6 entropy-23-01431-t006:** The sensitivity of key parameters mismatch.

$\left|\Delta a_{11}\right|\propto 10^{-13}$	$\left|\Delta a_{12}\right|\propto 10^{-13}$	$\left|\Delta a_{13}\right|\propto 10^{-13}$	$\left|\Delta a_{14}\right|\propto 10^{-13}$
$\left|\Delta a_{21}\right|\propto 10^{-13}$	$\left|\Delta a_{22}\right|\propto 10^{-13}$	$\left|\Delta a_{23}\right|\propto 10^{-13}$	$\left|\Delta a_{24}\right|\propto 10^{-13}$
$\left|\Delta a_{31}\right|\propto 10^{-13}$	$\left|\Delta a_{32}\right|\propto 10^{-13}$	$\left|\Delta a_{33}\right|\propto 10^{-13}$	$\left|\Delta a_{34}\right|\propto 10^{-13}$
$\left|\Delta a_{41}\right|\propto 10^{-13}$	$\left|\Delta a_{42}\right|\propto 10^{-13}$	$\left|\Delta a_{43}\right|\propto 10^{-13}$	$\left|\Delta a_{44}\right|\propto 10^{-13}$

**Table 7 entropy-23-01431-t007:** PSNR and SSIM test results.

Original and Encrypted Images	PSNR	SSIM
P1&E1	6.5326 dB	0.0411
P2&E2	6.2433 dB	0.0375
P3&E3	6.5768 dB	0.0314

## Data Availability

Not applicable.

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
