# Peer review of "Design and ARM-Based Implementation of Bitstream-Oriented Chaotic Encryption Scheme for H.264/AVC Video"

_entropy, 2021, doi:10.3390/e23111431_

Round 1

Reviewer 1 Report

The idea proposed by the authors is to improve bitstream-oriented encryption (BOE) based on chaotic encryption for H.264/AVC video.  This idea is very good and very promising to ensure security, format compliance and real-time transmission of encrypted videos. The paper is very well written and well structured. 

My only concern is related to the comparison of the results with only one literature method. It is recommended that you have more references in the literature to compare the results with.

Author Response

Response to Reviewer 1 Comments

Comment 1: The idea proposed by the authors is to improve bitstream-oriented encryption (BOE) based on chaotic encryption for H.264/AVC video. This idea is very good and very promising to ensure security, format compliance and real-time transmission of encrypted videos. The paper is very well written and well structured.

Reply: Thank you very much for your recognition of our articles and work, which is very important to us.

Comment 2: My only concern is related to the comparison of the results with only one literature method. It is recommended that you have more references in the literature to compare the results with.

Reply: Thank you very much for your advice. In the related research work in the past decade, there are few references studied on the hardware design and optimization of video chaotic secure communication. According to our survey and analysis, there are five references related to hardware design and chaos. The comparative analysis is as follows:

  • Reference [1] implemented video encryption on ARM-embedded platform. The video encryption scheme of this reference is: RGB video is encrypted before PNG compression. Therefore, the video encryption scheme belongs to full encryption method. The encrypted video format is PNG, which meets the video format compatibility. However, this scheme has the shortcomings of large amount of encrypted data and low efficiency, so the system is far from meeting the real-time requirements of 25  In addition, no specific transmission frame rate is given in this reference, but we have reproduced and tested the system frame rate to be about 5 fps.
  • Reference [2] is the continuation of the work of the reference [1]. In order to improve the real-time performance of the system, the encrypted H.264 video is also encrypted by full encryption (FE) method in the same ARM-embedded platform. Because the data volume of one frame of H.264 video is much smaller than that of a frame of RGB video, and the video transmission frame rate of the system is as high as 26.8 fps by using the multi-core and multi-threaded optimization method. The real-time performance of the system meets the requirements, but the H.264 video format is damaged and does not meet the requirements of format compatibility. However, Format compatibility is an important indicator of video chaotic secure communication system, so the scheme of reference [2] still has some drawbacks.
  • In order to solve the contradiction between real-time and format compatibility pointed out above, based on the same ARM-embedded platform, the reference [3] proposed a selective encryption method, which transplants chaotic encryption algorithm into H.264 software encoding library. The H.264 video is encrypted during the encoding and compression process, and finally the encrypted video with 264 format is output. Although this scheme can keep the encrypted video format compatible, the real-time performance is still poor due to the heavy load on the ARM core caused by the high computation consumption of the software encoder. Reference [3] does not directly give the frame rate test data, but after our implementation and experimental tests, the frame rate is about 17 fps, which has better real-time performance than reference [1], but still cannot meet the requirements of 25 fps. The BOE method proposed in our paper is a further optimization and improvement of reference [3]. It uses hardware codec instead of software codec, which greatly reduces the load of ARM core and is conducive to improving system efficiency. The scheme uses both entropy decoding and entropy encoding to keep the format compatibility of encrypted video and achieve video transmission frame rate of 27 fps. The scheme proposed in our paper is the best one with format compatibility and real-time performance in current related research work.
  • Reference [4] proposes to use Xilinx ZYNQ 7000 series SoC architecture to achieve video chaos secure communication. Using hardware-software co-design method, RGB video is encrypted and video communication is implemented, and video transmission frame rate of 31 fps is obtained. However, the scheme encrypts RGB video directly without video encoding compression, it does not conform to the video communication specification.
  • Reference [5] is an extension of the work of reference [4]. Using Xilinx Virtex 7 type of FPGA chip, the system functions of reference [4] are all implemented by FPGA, which achieves a higher system frame rate of 67.8 fps. However, the scheme still has not conform to the video communication specification.

To sum up, reference [2] was added to the revised paper as a new comparative reference (Pages 20-21 of the revised paper). Reference [1] cannot be added for comparison because it does not have frame rate test data; References [4] and [5] lack comparability because they use FPGA architectures rather than ARM architectures.

REFERENCE

[1] Lin Z, Yu S, Lü J, et al. Design and ARM-embedded implementation of a chaotic map-based real-time secure video communication system[J]. IEEE Transactions on circuits and systems for video technology, 2014, 25(7): 1203-1216.[2] Chen G . Control and anticontrol of chaos[M].  

[2] Chen P, Yu S, Zhang X, et al. ARM-embedded implementation of a video chaotic secure communication via WAN remote transmission with desirable security and frame rate[J]. Nonlinear Dynamics, 2016, 86(2): 725-740.

[3] Zhang X, Yu S, Chen P, et al. Design and ARM-embedded implementation of a chaotic secure communication scheme based on H. 264 selective encryption[J]. Nonlinear Dynamics, 2017, 89(3): 1949-1965.

[4] Chen P, Yu S, Chen B, et al. Design and SOPC-based realization of a video chaotic secure communication scheme[J]. International Journal of Bifurcation and Chaos, 2018, 28(13): 1850160.

[5] Chen B, Yu S, Chen P, et al. Design and virtex-7-based implementation of video chaotic secure communications[J]. International Journal of Bifurcation and Chaos, 2020, 30(05): 2050075.

Reviewer 2 Report

Dear authors and editor

I tried to carefully read the work but to me the methods are not adequately described 

I had trouble understanding the decryption and encryption procedure. I am not sure what are  the key values that the receiver should know and and how the receiver decrypts the informationalso thr chaotic map used is not adequately analysed. The parameter values are very different from each other, some being on scale of 10^10 and others much too specific,  and on scale of of 10^-5. This to me is much too specific for obtaining a clear scope of the map's behavior. Also there are no bifurcation diagrams of the map so it is impossible to get a clear view of how the map behaves if a single parameter changes.

Also,. Note that feeding an external signal to the map may affect its chaotic behavior.

Also in 11,. I am not sure how u mean that the xor is performed between the two bytes and then fed into the system? Do you perform xor on the binary representation and then transform the result back into an integer?

Reviewer 3 Report

The peer-reviewed article is a thorough study dedicated to solving an urgent problem. Chaotic encryption is a relevant application of the theory of nonlinear dynamics. However, I cannot recommend the article for publication in its current form due to the following reasons:

1. The article includes a quite detailed state of the art. However, from the introduction, it is unclear why algorithms based on chaotic systems should be used for encryption for H.264 / AVC Video.
2. The introduction says that 4-D SCSCA-CAC encryption can effectively resist the cryptanalysis method combining a chosen-ciphertext attack with a divide-and-conquer attack and pass the TESTU01. In my opinion, it is worth reformulating this statement since the TestU01 is primarily aimed at testing the hypothesis of the randomness of bit sequences and not breaking the encryption algorithm.
3. Some other analyzes are required, including NIST testing as well as a phase space reconstruction attack.
4. It would be interesting to see the results of comparing the speed of the proposed encryption algorithm with other chaos-based methods and traditional approaches. For example, it has recently been shown that encryption based on the adaptive chaotic map performs faster than the classical AES algorithm.

Round 2

Reviewer 3 Report

Dear Authors,

Thank you for taking my recommendations into account. I recommend the article for publication as it is.